# A MTA2-SATB2 chromatin complex restrains colonic plasticity toward small intestine by retaining HNF4A at colonic chromatin

Wei Gu [1,6,7] ✉, Xiaofeng Huang [1,7], Pratik N. P. Singh[2,3], Sanlan Li[1], Ying Lan[1], Min Deng[1], Lauretta A. Lacko[1,4], Jesus M. Gomez-Salinero [1], Shahin Rafii [1], Michael P. Verzi [5], Ramesh A. Shivdasani [2,3] & Qiao Zhou [1,4] ✉

Plasticity among cell lineages is a fundamental, but poorly understood, property of regenerative tissues. In the gut tube, the small intestine absorbs nutrients, whereas the colon absorbs electrolytes. In a striking display of inherent plasticity, adult colonic mucosa lacking the chromatin factor SATB2 is converted to small intestine. Using proteomics and CRISPR-Cas9 screening, we identify MTA2 as a crucial component of the molecular machinery that, together with SATB2, restrains colonic plasticity. MTA2 loss in the adult mouse colon activated lipid absorptive genes and functional lipid uptake. Mechanistically, MTA2 co-occupies DNA with HNF4A, an activating pan-intestinal transcription factor (TF), on colonic chromatin. MTA2 loss leads to HNF4A release from colonic chromatin, and accumulation on small intestinal chromatin. SATB2 similarly restrains colonic plasticity through an HNF4A-dependent mechanism. Our study provides a generalizable model of lineage plasticity in which broadly-expressed TFs are retained on tissue-specific enhancers to maintain cell identity and prevent activation of alternative lineages, and their release unleashes plasticity.

Cells in regenerative tissues can exhibit substantial phenotypic plasticity upon injury[1-3]. Differentiated quiescent cells may dedifferentiate along its lineage trajectory to become progenitors or stem cells, then assume forward differentiation to produce progenies for tissue repair, as reported in the lung, intestine, and skin[4-7]. Cells could also cross lineage boundaries and switch fate. Examples of lineage plasticity include conversion of hepatocytes to cholangiocytes in the liver, alpha/delta to beta cells in the pancreas, and hair follicle stem cells to epidermal stem cells in the skin[8-12]. Lineage plasticity must be tightly regulated because undesirable plastic events, such as metaplasia, may promote dysfunction or serve as precursors to tumorigenesis[13,14].

Some of the signaling pathways, transcriptional mediators, and chromatin substrates of cellular plasticity have begun to emerge from recent studies[8,15-18]. Nevertheless, our understanding of the molecular complexes that safeguard cellular identity and mediate lineage plasticity remains limited.

The small and large intestines are markedly different in cell composition and function[19]. While the small intestine absorbs nutrients through the enterocytes that line the mucosal surface, the colon mainly absorbs electrolytes and water. Accordingly, enterocytes express many transporters for lipids, amino acids, and carbohydrates that the colonocytes lack[20]. Nevertheless, using genome-wide mapping

[1]Division of Regenerative Medicine & Hartman Institute for Organ Regeneration, Department of Medicine, Weill Cornell Medicine, 1300 York Avenue, New York, NY 10065, USA. [2]Department of Medical Oncology, Center for Functional Cancer Epigenetics, Dana-Farber Cancer Institute, 450 Brookline Avenue, Boston, MA 02215, USA. [3]Department of Medicine, Brigham and Women's Hospital, Harvard Medical School, 75 Francis Street, Boston, MA 02115, USA. [4]Human Therapeutic Organoid Core Facility, Weill Cornell Medicine, 1300 York Avenue, New York, NY 10065, USA. [5]Department of Genetics, Rutgers University, 145 Bevier Road, Piscataway, NJ 08854, USA. [6]Present address: BeiGene Institute, BeiGene (Shanghai) Research & Development Co., Ltd, Shanghai 200131, China. [7]These authors contributed equally: Wei Gu, Xiaofeng Huang. ✉e-mail: wei.gu@beigene.com; jqz4001@med.cornell.edu

with the enhancer marker H3K4me1 and ATAC-seq (Transposase Accessible Chromatin with high throughput sequencing), colonic epithelial cells surprisingly harbor primed small intestine enhancers, suggesting an intrinsic permissiveness for small intestine gene activation in colon[21]. Upregulation of small intestine genes has been reported in the colon of patients with short bowel disease[22], whereas a more substantial colon to small intestine transcriptomic shift may occur in some patients with inflammatory bowel diseases (IBD)[23]. The colon thus displays various degrees of cellular plasticity.

We recently identified the colon-restricted chromatin factor, SATB2, as a central regulator of colonic gene expression and lineage plasticity[24]. *Satb2* deletion converts colonic stem cells into small intestine ileum-like stem cells, leading to the replacement of colonic mucosa by ileum-like mucosa in the adult mouse colon. Loss of SATB2 in mature colonocytes also rapidly activates ileal genes. Despite these observations, mechanisms of colonic plasticity remain largely unknown.

We hypothesized that some SATB2-associated chromatin factors may regulate colonic plasticity. Using Affinity Purification and Mass Spectrometry (AP-MS), we identified SATB2-associated proteins in colonic epithelium, including MTA2 (metastasis associated protein 2) and multiple members of the NuRD (Nucleosome Remodeling Deacetylase) complex. CRISPR (clustered regularly interspaced short palindromic repeats)-mediated loss-of-function evaluation in colonic organoids suggests that MTA2-containing NuRD complex could regulate colonic transcription. MTA2 expression is restricted to colonocytes. *Mta2* deletion in the adult mouse colon activated small intestine genes, including fat absorptive genes that enable lipid uptake by colonocytes. Mechanistic studies indicate that MTA2 and NuRD do not directly bind and silence small intestine genes, but rather MTA2 and the intestinal TF HNF4A, extensively co-occupied colonic chromatin. MTA2 loss led to HNF4A depletion at colonic enhancers and enrichment at small intestine enhancers, indicating a critical function for MTA2 in retaining HNF4A at colonic enhancers. Moreover, MTA2 physically interacts with SATB2, and both restrain HNF4A on colonic chromatin, albeit at different strength, leading to different degrees of plasticity in *Mta2* vs. *Satb2* mutant colon. Consistent with these findings, *Hnf4a* deletion from *Mta2*-null or *Satb2*-null colonic organoids abrogated small intestine gene activity. MTA2 loss led to stronger interaction of SATB2 with the NuRD core subunits HDAC1 and HDAC2, and HDAC1/2 inhibition suppressed small intestine genes, suggesting that enhanced HDAC1/2 activities near SATB2 may weaken HNF4A binding and activate small intestine genes. Together, our data reveal that a MTA2-SATB2 chromatin complex at colonic enhancers performs the dual functions of safeguarding transcriptional fidelity and regulating cell plasticity. By restraining HNF4A, a TF expressed in both small and large intestines, at colonic enhancers, transcription of small intestine genes is blocked in the colon and release of this block unleashes cellular plasticity.

## Results

### MTA2-containing NuRD associates with SATB2 and regulates colonic gene expression

SATB2 and its homolog SATB1 have been proposed as chromatin hubs that orchestrate protein-protein and protein-DNA interactions[25,26]. Reasoning that some of the SATB2-associated factors may regulate colonic plasticity, we purified protein complexes that contain SATB2 from murine colonic glands (Fig. 1a and Figure. S1a). Two independent AP-MS experiments identified a total of 628 proteins with a false discovery rate (FDR) < 1%. Of these, 78 proteins were significantly enriched in both samples (SATB2 AP-MS signal intensity and MS count over IgG controls > 2-fold), with SATB2 itself being the most enriched (Fig. 1b, Supplementary Data 1).

The top 40 candidate SATB2-associated proteins included an abundance of histones, nuclear matrix proteins, and chromatin remodeling factors, consistent with the proposed role of the SATB family as chromatin organizers (Fig. 1b and Figure. S1b). Four members of the NuRD complex, including CHD4, MTA2, RBBP4, and GATAD2A, were among the top 40 interactors, suggesting association of the NuRD complex with SATB2 (Fig. 1b). Using co-immunoprecipitation (co-IP), we observed interaction of both SATB2 with MTA2, and MTA2 with the NuRD core subunit CHD4 (Fig. 1c and Figure. S1c). We also validated the interaction of SATB2 with SMARCD2 and SMARCA4, two members of the SWI/SNF chromatin remodeling complex identified in AP-MS (Figure. S1c).

To evaluate the functional importance of candidate SATB2-associated factors in colonic transcription, we used CRISPR-CAS9 in murine colonic organoids to disrupt nine chromatin remodeling genes whose protein products were enriched in our AP-MS analysis (Figure. S1d); of these, seven achieved deletion efficiencies of 80-95% by immunoblot analysis (Fig. 1, e and Figure. S1e). RNA-sequencing indicated that deletion of *Chd4*, *Mta2*, or *Gatad2a*, but not the other factors, significantly altered colonic transcriptomes toward that of *Satb2* knockout organoids (Fig. 1f–h and Figs. S1F and 1g, Supplementary Data 2). These data suggest that the NuRD complex interacts with SATB2 and is functionally important in regulating colonic transcription.

### Activation of lipid absorptive genes in MTA2-deficient colonocytes

The colonic mucosa is a regenerative epithelium, with 4- to 7-day cycles of self-renewal powered by LGR5⁺ intestinal stem cells (ISCs) in the crypts of Lieberkuhn[27]. The colonic ISCs produce progenitors (transient amplifying cells) which give rise predominantly to absorptive colonocytes and secretory goblet cells (Fig. 2a). Immunohistochemistry revealed prominent MTA2 expression in upper, but not lower colonic glands, and in scattered sub-epithelial cells (Figs. 2b, c and Figure. S2a). In contrast, the NuRD subunits CHD4 and GATAD2A were present throughout the colonic epithelium, similar to SATB2 (Fig. 2b). The majority of MTA2⁺ cells (68.0 ± 7.6%) were CA1⁺ colonocytes and conversely, nearly all CA1⁺ mature colonocytes were MTA2⁺ (Figs. 2d and g). A minority subpopulation of MTA2⁺ cells were goblet cells (6.5 ± 3.1%), recognized by Alcian blue staining (Figs. 2f, g). LGR5⁺ colonic stem cells did not express MTA2 (Figs. 2e and g). Thus, an MTA2-containing NuRD complex is enriched in terminally differentiated colonocytes on the luminal surface.

Intestinal mucosa specific *Mta2* gene deletion in 2-month old *Villin-Cre^ER; Mta2^{f/f}* mice (Fig. 3a, hereafter referred to as Mta2^{cKO}), led to near complete absence of MTA2 (Fig. 3b). One month after Tamoxifen treatment, Mta2^{cKO} mice showed no overt changes in colonic histology or cell proliferation (Figure. S3a). RNA-sequencing of colonic glands revealed 200 up-regulated and 68 down-regulated genes (log₂ fold change [LFC] > 1, adjusted p [padj] <0.05) (Fig. 3c, Supplementary Data 3) in Mta2^{cKO} vs. control colon. Mta2^{cKO} colon was enriched for functional pathways and gene sets in fat digestion and absorption, thiamine metabolism, and chemokine signaling (Fig. 3d-f). Transporters for amino acids, carbohydrates, bile salts and vitamins also exhibited a trend toward up-regulation in Mta2^{cKO} colon (Figure. S3b). In contrast, no pathway was significantly enriched among the down-regulated genes (P < 0.001) (Fig. 3d). Immunohistochemistry showed expression of FABP6 and MTTP, two lipid transport proteins, in the upper glands of Mta2^{cKO} colon (Figs. 3g, h). Alkaline phosphatase, a small intestine brush border enzyme, was also activated and localized to surface colonocytes (Fig. 3g). Consistent with these molecular changes, BODIPY staining showed lipid accumulation in ileal villi and the upper glands of Mta2^{cKO} proximal colon, but not in control colon (Fig. 3i). Thus, MTA2 loss in colonocytes activated many small intestinal genes, particularly those involved in lipid transport and metabolism.

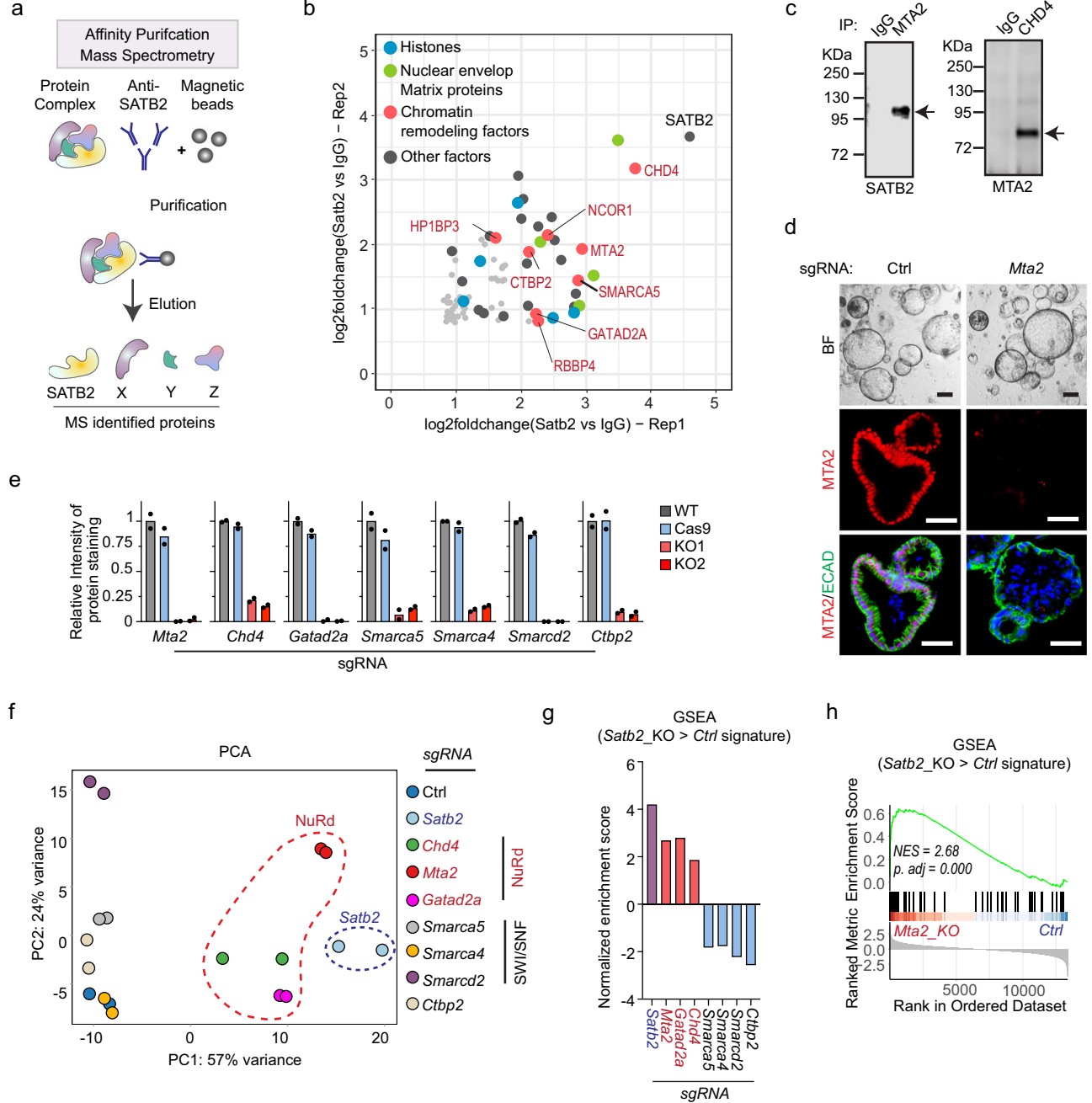

**Fig. 1 | MTA2 and the NuRD complex associate with SATB2 and regulate colonic transcription. a** Candidate SATB2-associated proteins were identified from mouse colonic glands by affinity purification (AP) with an anti-SATB2 antibody followed by Mass Spectrometry (MS). **b** Of the 78 proteins enriched in both AP-MS experiments, the top 40 (highlighted as colored balls) contained many histones, matrix proteins, and chromatin remodeling factors. **c** Co-IP demonstrated interactions of SATB2 with MTA2, and MTA2 with CHD4, a core member of the NuRD complex. Three independent experiments were repeated with similar results. **d** CRISPR-CAS9 and gRNA were used to successfully disrupt *Mta2* from cultured mouse colonic organoids, as shown by immunofluorescence staining. BF: bright field. ECAD: E-cadherin. Two independent experiments were repeated with similar results. Scale bar = 50 μm. **e** Immunoblot quantification showed significant reduction of seven SATB2-associated chromatin factors after CRISPR-mediated deletion in colonic organoids. Two independent CRISPR experiments and two controls were shown. Mean ± S.D. Source data are provided as a Source Data file. RNA-sequencing showed that disrupting NuRD members (*Chd4, Mta2* and *Gatad2a*) but not SWI/SNF factors (*Smarca5, Smarca4* and *Smarcd2*) or *Ctbp2* caused transcriptomic shifts of colonic organoids toward that of *Satb2* knockout, as illustrated by Principal Component Analysis (PCA, **f**) and Gene Set Enrichment Analysis (GSEA, **g**, **h**). NES: normalized enrichment score. *P* value was calculated by a phenotype-based permutation test and adjusted by Benjamini-Hochberg method. **g** Source data are provided as a Source Data file.

## MTA2 retains HNF4A on colonic enhancers and prevents HNF4A from activating small intestine chromatin

MTA2 is part of the NuRD complex, which has been proposed to suppress alternative transcriptional programs in several tissues by direct binding and suppression of target genes[28]. To investigate how MTA2 modulates small intestine gene expression in the colon, we mapped genome-wide MTA2 binding by chromatin immunoprecipitation sequencing (ChIP-seq). Duplicate MTA2 ChIP data from colonic epithelia yielded highly concordant data with 23,557 peaks (q < 1 ×10⁻³, using input DNA and Mta2^cKO as controls) (Figure. S4a). Colonic MTA2 binding occurred at promoters (49.1%, <2 kb from transcription start sites (TSSs)) and distal elements (50.9%, introns and intergenic

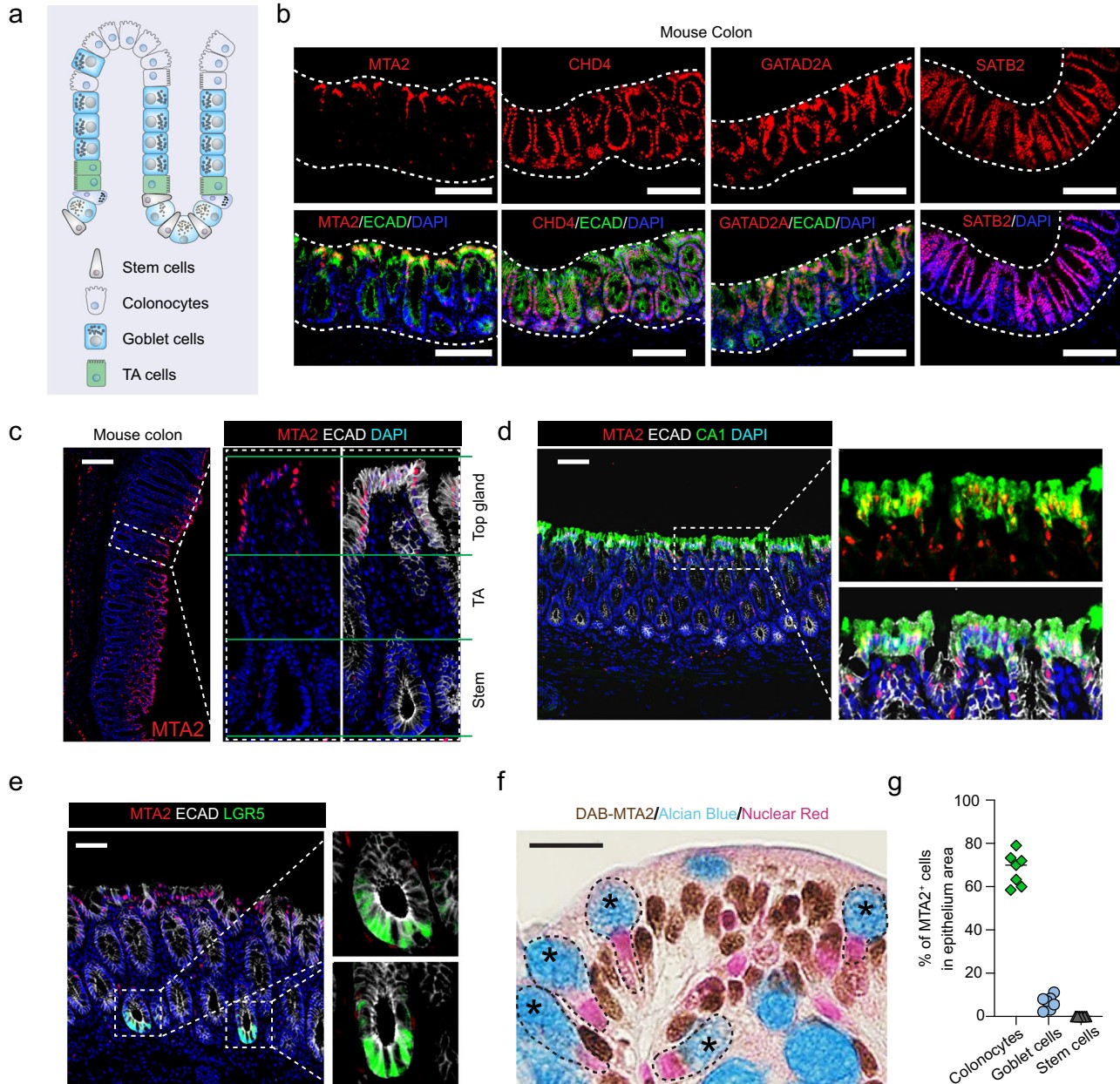

**Fig. 2 | MTA2 expression is enriched in colonocytes. a** Diagram of colonic epithelium and the cell lineages. Mature colonocytes are concentrated in the upper glands of mucosal surface. TA cells: transient amplifying cells.
**b** Immunofluorescence staining showed prominent MTA2 expression in the upper glands of adult mouse colon whereas SATB2 and two other members of NuRD, CHD4 and GATAD2A, were present throughout the epithelium. Two independent experiments were repeated with similar results. Scale bar = 100 μm. MTA2 was

expressed in Ecadherin⁺CA1⁺ mature colonocytes (**c**, **d**), but not in LGR5⁺ colonic stem cells (**e**). Seven independent experiments were repeated with similar results. Scale bar = 100 μm. **f** Histology stain for MTA2 and Alcian Blue stain for goblet cells showed few goblet cells expressing MTA2 (**f**). DAB: 3,3'-diaminobenzidine. Seven independent experiments were repeated with similar results. Scale bar = 25 μm. **g** MTA2 was present mostly in colonocytes. $n = 7$ independent experiments with 3 mice. Source data are provided as a Source Data file.

regions) (Figure. S4b). Genes near MTA2 binding sites (<50 kb) were highly enriched for the colonic but not the small intestine signatures (Fig. 4a, Supplementary Data 4). For instance, small intestine genes activated in *Mta2* null colon, such as *Lgals2*, *Npc1l1*, *Abcg8*, and *Pla2g2a* (Figs. 3c and f), lacked nearby MTA2 binding (Figure. S4c). These data indicate that MTA2 does not directly bind and suppress small intestine genes.

Using HOMER analysis, we identified the DNA-binding motif of the intestinal transcription factor HNF4A as the top enriched motif at distal MTA2 binding sites (Fig. 4b). HNF4A and its homolog HNF4G are expressed in both small and large intestines, and shown to be important in activating enterocyte gene transcription[29]. Thus,

we evaluated whether MTA2 could regulate HNF4A in the colon. Indeed, 87.2% of the MTA2 binding sites on colonic chromatin overlapped with binding of HNF4A (Fig. 4c). HNF4A expression was unchanged after MTA2 loss but HNF4A binding was depleted at 2,065 sites and acquired at an additional 4379 sites in Mta2^cKO colon (log2FC > 1.0, $q < 0.01$) (Figs. 4d, e and S4d). About 80% of depleted sites (1639 of 2065) and nearly all gained sites (4233 of 4379) were in distal elements (Figure. S4e), indicating that MTA2 regulates HNF4A binding primarily at distal enhancers. Consistent with this notion, the depleted and gained HNF4A sites corresponded to areas of open chromatin enriched in the colon and ileum, respectively (Fig. 4d). Moreover, loss and gain of HNF4A binding in Mta2^cKO colon were

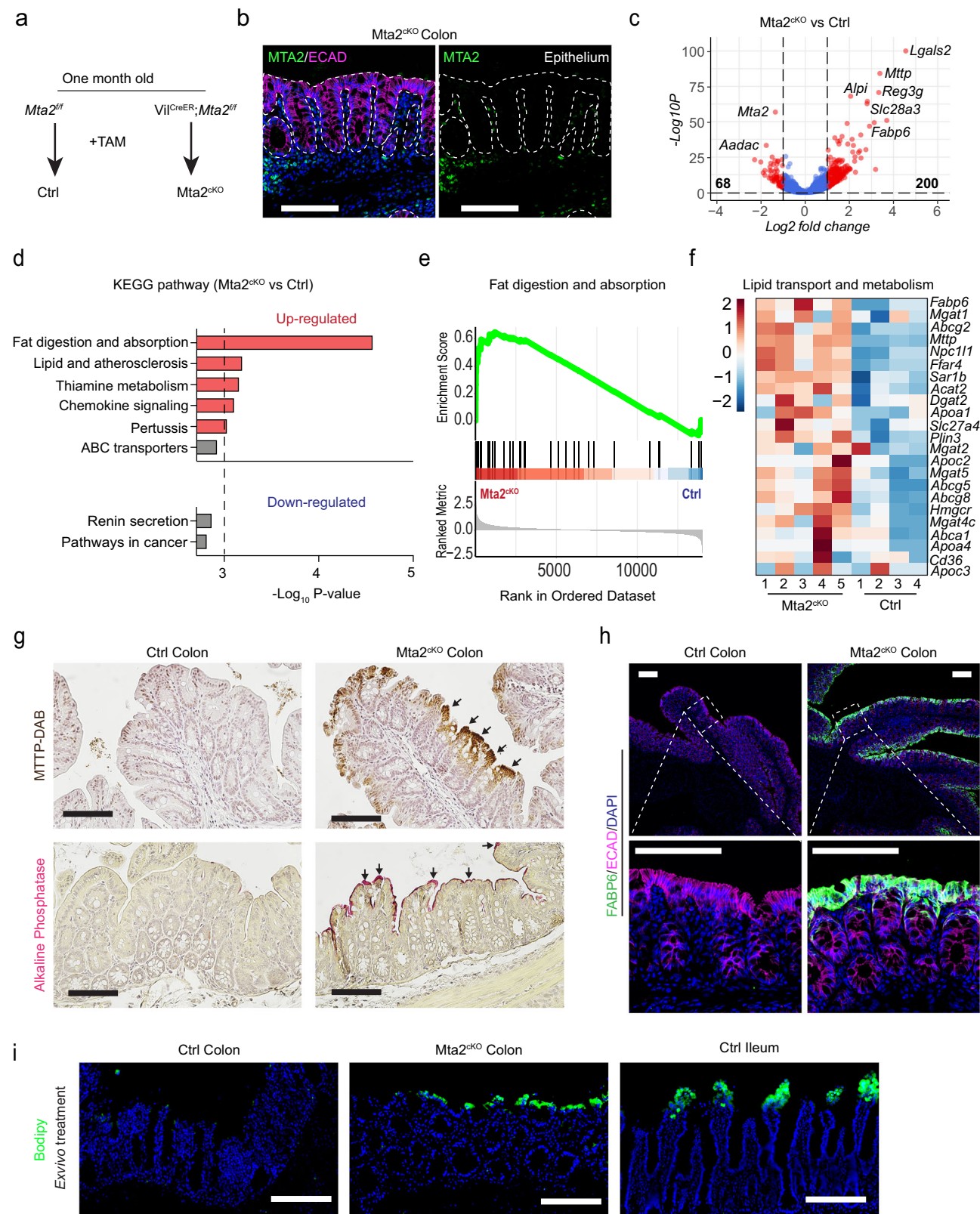

strongly associated with down-regulation of colonic and up-regulation of ileal genes, respectively (Figs. 4f, g and S4F, S4g). Thus, MTA2 deletion led to HNF4A loss on colonic enhancers and its relocation to small intestine enhancers, triggering activation of small intestine genes in the colon. These data suggest that MTA2 retains HNF4A binding on colonic chromatin and prevents HNF4A from activating small intestine genes.

## Both SATB2 and MTA2 co-localize with HNF4A on colonic chromatin but SATB2 restrains HNF4A more strongly than MTA2

Both SATB2 and MTA2 can regulate colonic plasticity and our findings indicate that they interact physically (Fig. 1a-c). Structural studies of SATB1/2 proteins have identified five functional domains: a N-terminal ubiquitin-like domain (ULD) that mediates oligomerization, a CUT-like

**Fig. 3 | Activation of lipid transport and metabolism genes in adult mouse colon after MTA2 loss.** *Mta2* was deleted from 2-month old *Vil^CreER^;Mta2^f/f^* (Mta2^cKO^) mice by applying tamoxifen (**a**), leading to near complete absence of MTA2 in colonic epithelium (**b**). Three independent experiments were repeated with similar results. Scale bar = 100 μm (**b**). RNA-seq of control and Mta2^cKO^ colonic glands identified 200 up-regulated and 68 down-regulated genes ([LFC] > 1, adjusted p [padj] <0.05) (**c**, volcano plot). *P* value calculated by Wald test and adjusted by Benjamini-Hochberg method. Although no molecular pathways were significantly enriched among the down-regulated cohort, genes involved in lipid absorption, transport, and metabolism were prominently enriched among the up-regulated cohort, as illustrated by KEGG pathway gene set enrichment analysis (**d**, **e**) and in the heatmap representation (**f**). *P* value calculated by a phenotype-based permutation test and adjusted by Benjamini-Hochberg method. **d** Source data are provided as a Source Data file. **g**, **h** Histology and immunofluorescence staining showed activation of lipid transport proteins FABP6 and MTTP and small intestine brush border enzyme Alkaline Phosphatase in the surface colonocytes of Mta2^cKO^ colon. Three independent experiments were repeated with similar results. Scale bar = 100 μm. **i** BODIPY stain revealed presence of lipid accumulation in villi of ileum and surface glands of Mta2^cKO^ proximal colon, but not control colon. Two independent experiments were repeated with similar results. Scale bar = 100 μm.

domain (CUTL) and two CUT domains (CUT1 and CUT2) that are critical for DNA binding, and a c-terminal HOX domain[30] (Fig. 5a). Although HOX domains often serve as a primary DNA binding domain, this is not the case for SATB1/2[31]. The primary function of the HOX domain in SATB1/2 is unclear. We generated five SATB2 mutant proteins, each lacking one of the five functional domains (Fig. 5a). Co-IP studies with the mutant SATB2 proteins revealed that MTA2 interacts with SATB2 primarily via the HOX domain (Fig. 5b).

The physical interaction of MTA2 and SATB2 suggests that both proteins might co-localize with HNF4A on colonic chromatin. Because both MTA2 and SATB2 regulate HNF4A primarily at distal genomic sites (this study and reference 21), we assessed distal co-localization of the three factors. Alignment of MTA2 peaks with published SATB2 ChIP data showed 44.5% co-occupancy at distal elements, whereas 36.8% of distal MTA2 sites were co-bound by both SATB2 and HNF4A (Figs. 5c, d). Despite this extensive co-localization, SATB2 loss in the colon activated more small intestine genes than MTA2 loss, and larger numbers of HNF4A binding sites were lost and gained in Satb2^cKO^ (*Villin-Cre^ER^; Satb2^f/f^*) than in Mta2^cKO^ colon. The lost and gained sites correspond to sites of open chromatin in the colon and ileum, respectively (Figs.5e, f, and 6a).

In *Satb2*-null colon, large numbers of MTA2 genomic binding sites shifted in parallel with those of HNF4A (Figs. 6a, b). In contrast, many fewer SATB2 binding sites were depleted or gained in *Mta2*-null colon (Fig. 7a) and these alterations were not associated with dysregulation of colonic or ileal genes (Figs. 7b, c). Thus, SATB2 regulates genomic binding of MTA2, but not vice versa. Altogether, these data imply that both MTA2 and SATB2 restrain HNF4A at colonic chromatin, but SATB2 regulates MTA2 and more robustly retains HNF4A binding.

## Small intestine gene activation in both Mta2^cKO^ and Satb2^cKO^ colon depends on HNF4A

Our chromatin mapping data implicated HNF4A as a key mediator of colon-small intestine plasticity. Gain of HNF4A binding on small intestine chromatin is tightly associated with transcriptional activation of small intestine genes in Mta2^cKO^ or Satb2^cKO^ colon (this study and reference 21). If this association is causal, then removing HNF4A should block colonic transcriptional plasticity. To evaluate this hypothesis, we differentiated murine colonic organoids into colonoids enriched for CA1^+^ colonocytes (Figs. 8a and b). We used CRISPR-Cas9 to delete *Hnf4a* from either Mta2^cKO^ or Satb2^cKO^ colonoids, achieving deletion efficiencies > 90% (Figs. 8c and e). qPCR analysis of representative small intestine genes showed that their activation was strongly attenuated in both mutants (Figs. 8d and f). Thus, small intestinal gene activation in both Mta2^cKO^ and Satb2^cKO^ colon depends on *Hnf4a*.

## Increasing HNF4A dosage in the colon activates small intestine gene transcription

Given that colonic enhancers are occupied by HNF4A whereas the small intestine enhancers are primed but lack HNF4A binding in colon, we reasoned that "excess" HNF4A, provided by ectopic expression, should engage small intestine enhancers in colon and activate transcription. To test this hypothesis, we over-expressed HNF4A in cultured mouse colonic organoids. RNA-seq showed 89 up-regulated and 7 down-

regulated genes (log$_2$ fold-change >1, *p* < 0.05, Figs. 9a, f). Gene sets characteristic of small intestine functions, such as cholesterol metabolism, fat and protein digestion and absorption, and retinol metabolism, were enriched among the up-regulated genes (Fig. 9b). We next over-expressed HNF4A in 5 independent human colonic organoid lines. RNA-seq studies revealed 90 up-regulated and only 2 down-regulated transcripts (log$_2$ fold-change >1, *p* < 0.05, Figs. 9c, d, f). Small intestine functional pathways, including fat, protein and carbohydrate digestion and absorption, were activated (Figs. 9e and f). Thus, loss- and gain-of-function studies implicate HNF4A as a mediator of small intestine gene activity in colonic plasticity, conserved across species.

## HDAC activity is required for small intestine gene activation in MTA2^cKO^ colon

We hypothesized that *Mta2* loss may lead to compositional and/or conformational changes of the SATB2-NuRD complex, resulting in altered HNF4A binding at colonic enhancers and activation of small intestine genes. Indeed, AP-MS of the SATB2 complex from *Mta2^cKO^* colon showed 71 enriched and 25 depleted proteins, compared with wild-type colon (signal intensity >2-fold or <2-fold in mutant vs. control samples, Supplementary Data 5). Immunoblots of colonoids showed less of the NuRD core subunit HDAC2 (*P* = 0.018) in *Mta2* mutants whereas other core subunits, CHD4 and HDAC1, were no different (Fig. 10a). Co-IP studies, however, revealed stronger interactions of SATB2 with both HDAC1 and HDAC2, but not with CHD4, after *Mta2* loss (Fig. 10b). Treatment of *Mta2* mutant colonoids with HDAC1/2 inhibitors 4-phenylbutyric acid (4PBA) and SAHA strongly attenuated expression of small intestine genes, including *Abcg8, Lgals2, Pla2g2a* and *Slc43a1* (Fig. 10c), suggesting that enhanced HDAC1/2 activities near SATB2 may weaken HNF4A binding and drive small intestine gene activation. The active enhancer mark H3K27ac was not reduced at genomic sites depleted of HNF4A in *Mta2^cKO^* colon, indicating that H3K27ac is not a primary target of HDAC1/2 at colonic enhancers (Figure. S5).

Collectively, our data support a model in which chromatin factors MTA2-NuRD and SATB2 form a complex to retain HNF4A at colonic chromatin and the degree of plasticity relates to the amount of HNF4A released from that retention. If relatively little HNF4A is liberated, as with MTA2 loss, then the increase in small intestine gene activity is modest; if more HNF4A is released, as occurs with SATB2 loss, then a larger transcriptomic shift ensues, with overt phenotypic tissue conversion (Fig. 10d).

## Discussion

To perform specialized functions, distinct cell types must maintain unique identities, including cell type-specific transcriptomes. They also need to adapt to changing environments by deploying plasticity in transcription or even cell identity[32–34]. The molecular control of cell identity and plasticity is a fundamental question, with scant mechanistic insights.

We previously uncovered surprising plasticity between the adult colon and ileum, controlled by the colon-restricted chromatin factor, SATB2. SATB2 deletion causes drastic cell fate switch, converting colonocytes to enterocytes. In this study, we identified MTA2 as a new component of the molecular machinery that preserves colonic identity

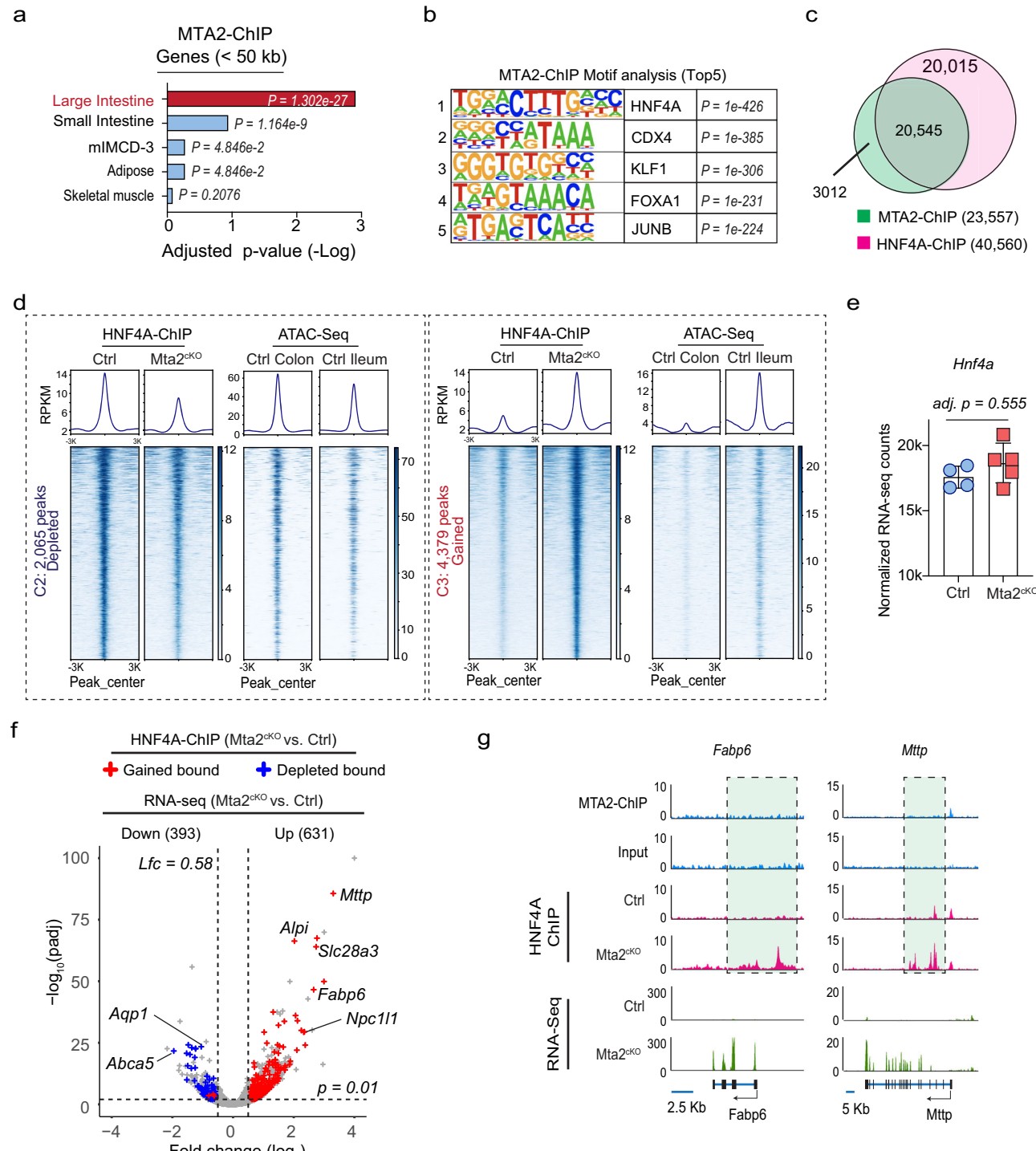

**Fig. 4 | MTA2 retains HNF4A on colonic enhancers and prevents HNF4A from activating small intestine chromatin. a** Tissue enrichment scores of genes near MTA2 binding peaks (MACS2, q < 0.001, distance <50 kb) in colon showed a predominant colonic signature. The *P* value was calculated using Enrichr tool which relies on Fisher's exact test and adjusted by Benjamini-Hochberg method. **b** Top 5 DNA binding motifs in MTA2 distal binding sites ranked by *P* value. The *P* value calculated using HOMER tool which used cumulative binomial distributions to calculate motif enrichment. **c** Venn diagram showing the overlap of MTA2 and HNF4A genomic binding in the colon. **d** DNA binding profiles of HNF4A sites that were either reduced (left panel, 2,065 sites) or gained (right panel, 4379 sites) after MTA2 loss in the colon. Corresponding ATAC signals in the colon or ileum are

shown. Plots are 6-kb windows centered on each HNF4A binding site. **e** HNF4A mRNA levels were comparable in Mta2^cKO vs. control colon. Mean ± S.D. *n* = 4 independent control mice and *n* = 5 independent Mta2^cKO mice. Adjusted *p* value by Wald test corrected for multiple testing with Benjamini and Hochberg method. Source data are provided as a Source Data file. **f** Combined RNA-seq and HNF4A ChIP plot showed that after MTA2 deletion, the increase (up) or decrease (down) of gene expression was strongly associated with gain or loss of HNF4A binding. Adjusted *p* value calculated by Wald Test from DESeq2. **g** Genome Browser tracks of MTA2 and HNF4A binding and RNA-seq at genomic loci of two small intestine lipid transport genes. Gain of HNF4A binding at these loci (highlighted) correlated with transcriptional activation.

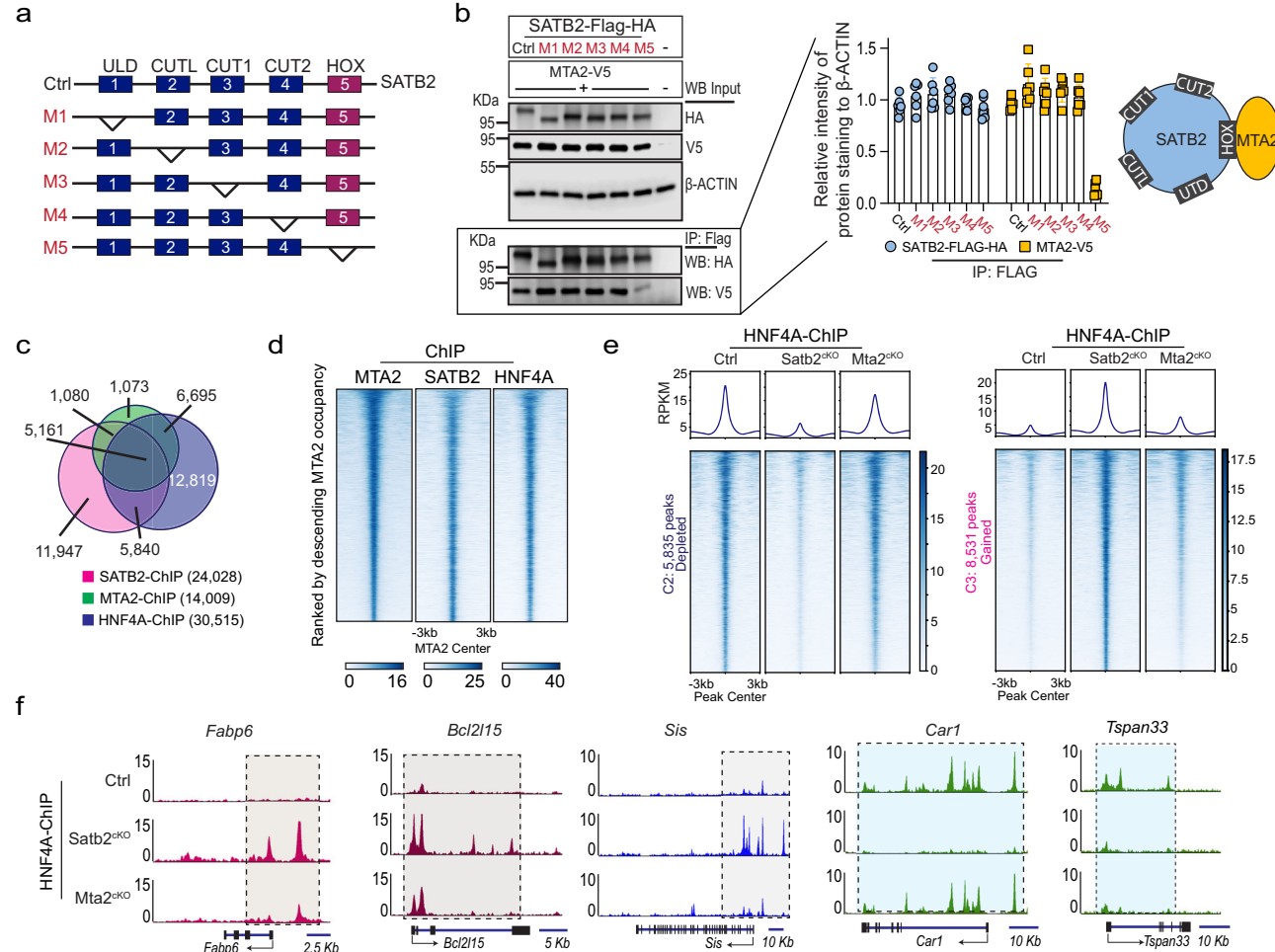

**Fig. 5 | SATB2 and MTA2 co-bind HNF4A on colonic chromatin but SATB2 retains HNF4A more strongly than MTA2. a, b** We generated 5 mutant SATB2 proteins (M1-5) with each lacking one of the 5 functional domains. **a** Co-IP of the SATB2 mutants and MTA2 showed that the SATB2 HOX domain was required for SATB2 interaction with MTA2; without the HOX domain, the interaction was abrogated. M: mutation form. **b** Mean ± S.D. $n = 6$. All the different gels/blots were derived from the same experiment and were processed in parallel. Source data are provided as a Source Data file. Overlap of SATB2, MTA2, and HNF4A distal genomic binding sites in colonic tissues as shown in the Venn diagram (**c**) and the DNA binding profiles (**d**). Peaks were ranked by descending MTA2 occupancy. **e** DNA binding profiles of HNF4A sites that were either reduced (left panel, 5835 sites) or gained (right panel, 8531 sites) in Satb2cKO colon. In comparison, HNF4A loss or gain at these sites were modest in Mta2cKO colon, but nonetheless statistically significant by Kolmogorov-Smirnov test (K-S D values shown in the density plots). Peaks centered on HNF4A binding sites in 6 kb windows. **f** Genome Browser tracks of HNF4A binding at genomic loci of the small intestine genes *Fabp6*, *Bcl2l15*, and *Sis* and the colonic genes *Car1* and *Tspan33* in Mta2cKO and Satb2cKO colon. SATB2 can more strongly influence HNF4A binding than MTA2.

and mediates plasticity. MTA2 loss led to activation of a subset of ileal genes and modest down-regulation of colonic genes. Thus, MTA2 and SATB2 regulate colonic plasticity to different degrees. Nevertheless, a common underlying mechanism is the ability of MTA2 and SATB2 to retain HNF4A on colonic chromatin. SATB2 appears able to strongly "tether" HNF4A to colonic enhancers, and thus its loss leads to transcriptome-wide changes. MTA2, however, tethers a fraction of HNF4A to colonic enhancers, and its loss causes modest transcriptional changes. Notably, small intestine gene activation in both MTA2 and SATB2 knockout colon is blocked by removing HNF4A, highlighting the key role of HNF4A in mediating colonic plasticity. SATB2 regulates genomic binding of both MTA2 and HNF4A, but not vice versa, highlighting the central importance of SATB2 in regulating colon-ileum plasticity.

MTA2 is part of the NuRD complex, which alters nucleosome spacing and deacetylates histones and other proteins[35–38]. Incapacitating NuRD in multiple tissues, including lymphoid cells and muscles, activates alternative lineage programs[39,40]. However, in these cases, NuRD is proposed to directly bind and silence alternative lineage genes, a mechanism distinct from the one we observed in colonic to

ileal plasticity. Our data suggest that MTA2 loss leads to conformational changes that bring HDAC1/2 of the NuRD complex closer to SATB2. These HDACs may then modify proteins around the SATB2 complex, making the local chromatin milieu less favorable for HNF4A binding. This notion is consistent with studies in embryonic stem cells, where NuRD binding at active enhancers is reported to modulate TF binding in different contexts[41], but NuRD targets on colonic chromatin have yet to be identified. In sum, our study reveals a chromatin complex that serves the dual purpose of preserving colonic identity while allowing plasticity. This model of cellular plasticity may apply broadly, that is, chromatin complexes restrain certain TFs at tissue-specific chromatin to prevent them from activating genes for alternative lineages and the measured release of these factors elicits different degrees of cellular plasticity.

## Methods
### Mouse strains
The MTA2loxp/loxp (MTA2f/f) strain was a gift from Dr. Robert G[42]. Roeder of the Rockefeller University and Dr. Yi Zhang of Harvard Medical School who originally made the strain. Vil-CreERT2 strain was a gift

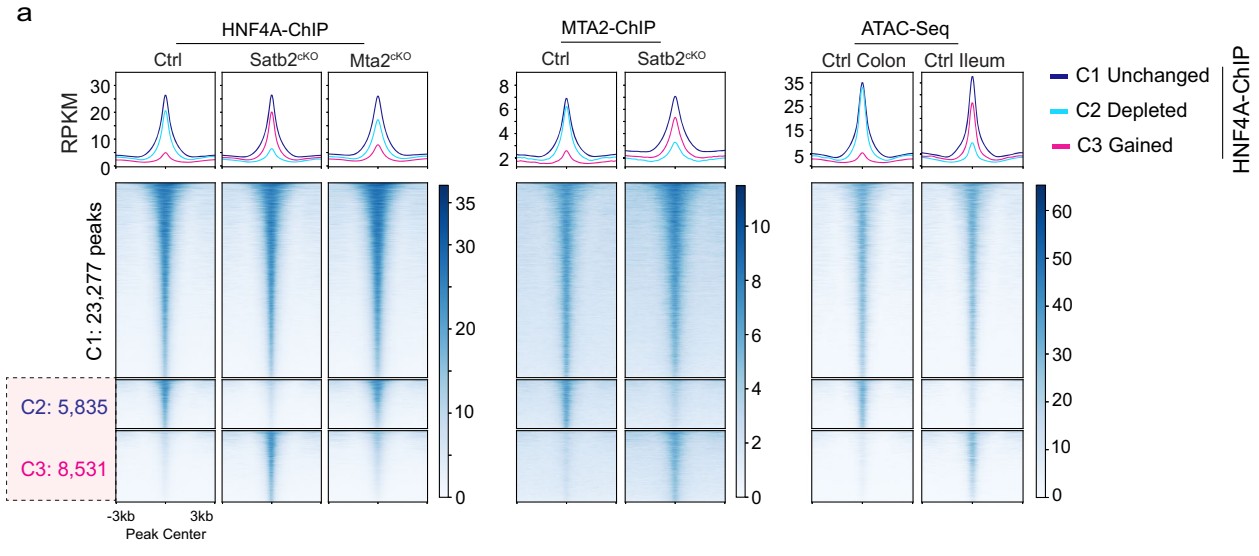

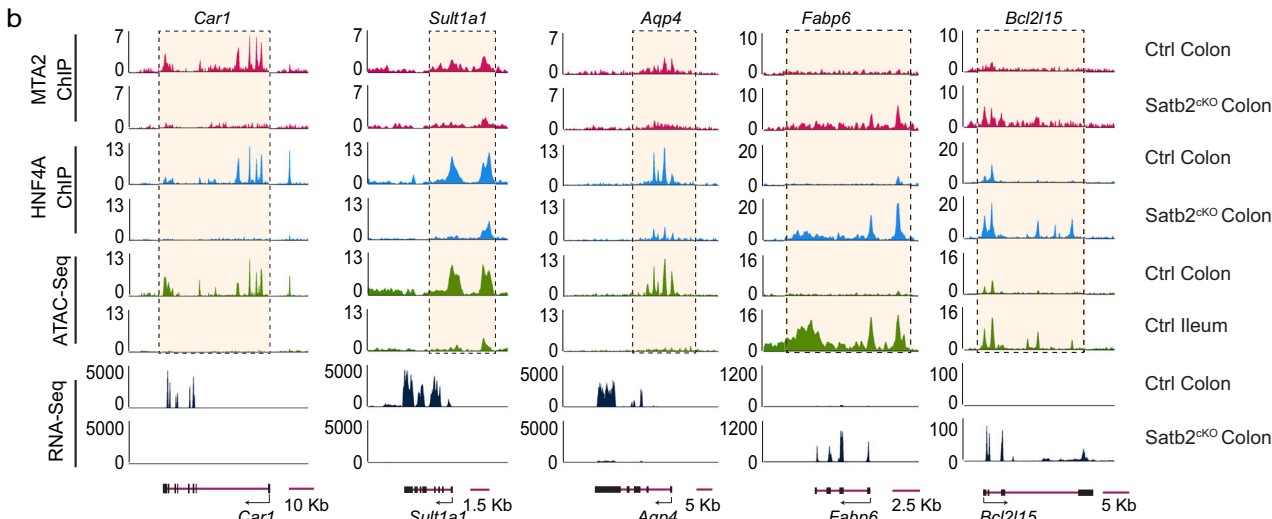

**Fig. 6 | SATB2 regulates genomic binding of both MTA2 and HNF4A. a** DNA binding profiles of HNF4A sites that were either reduced (C2, 5,835 sites) or gained (C3, 8,351 sites) in Satb2cKO colon. Corresponding MTA2 binding profiles and ATAC profiles in wild-type colon and ileum were shown. MTA2 binding loss and gain paralleled that of HNF4A in Satb2cKO colon, indicating that both MTA2 and HNF4A genomic binding were regulated by SATB2. Plots are 6-kb windows centered on each MTA2 binding site. **b** Genome Browser tracks showed coordinated shifts in MTA2 and HNF4A binding, ATAC and mRNA profiles at genomic loci of three colonic genes (*Car1, Sult1a1,* and *Aqp4*) and two small intestine genes (*Fabp6* and *Bcl2l15*) before and after *Satb2* deletion.

from Sylvie Robine (Institute Pasteur)[43]. To confer conditional deletion of MTA2, 4 mg per 25 g of body weight of tamoxifen (TAM, 10 mg per ml in corn oil) was intraperitoneally injected into both Mta2cKO (Vil-CreERT2; *Mta2*f/f) and Ctrl (*Mta2*f/f) mice (male and female equally, at 2 months old) once every 2 days for a total of 3 times. Mice were housed and bred in an ambient temperature (18 °C–22 °C) - and humidity (50–60%)- controlled environment with 12 hours light/dark cycle and food/water supplement.

## Mouse colonic crypt isolation, colonoid culture, and differentiation

Mouse colonoid line establishment and culture were performed as previously described with modification[21,44]. All the experiments were performed on ice or at 4 °C unless specified. Briefly, proximal colon top glands were scraped by glass slides and then the tissues were cut into approximately 0.3 cm size pieces and incubated in 10 mM EDTA for 30 min. The tissues were transferred to 15 mL cold PBS solution. After vigorous pipetting with 1% BSA pre-coated 10 mL serological

pipettes, epithelium cell clumps were collected by centrifugation at 300 $g$ for 5 min. Crypts were further isolated by filtering through a 70 μm cell strainer. 25–100 Crypts (P0) per 12 μl Matrigel™ droplet were cultured in WENR medium (Supplementary Table 1) in humidified chambers containing 5% $CO_2$ at 37 °C for 4–5 days with one-time medium change at day 2. After one passage, the P1 colonoids were differentiated into colonocyte-enriched colonic organoids by culturing in differentiation medium (DEM) (WENR medium without WRN conditioned medium and with the addition of 1 μg/mL RSpondin and 10 μM L-161,982, Supplementary Table 1) at day 2. Three days after differentiation, the organoids were either directly lysed in RNA lysis buffer (ZYMO) for RNA exaction, or incubated with cell recovery solution on ice, to remove Matrigel, for immunofluorescence, immunoblotting, and immunoprecipitation analyses.

## CRISPR-mediated gene knockout in colonoids

Murine *Mta2, Chd4, Gatad2a, Smarca4, Smarca5, Smarcd2* and *Ctbp2* sgRNAs were designed with the Synthego CRISPR design tool

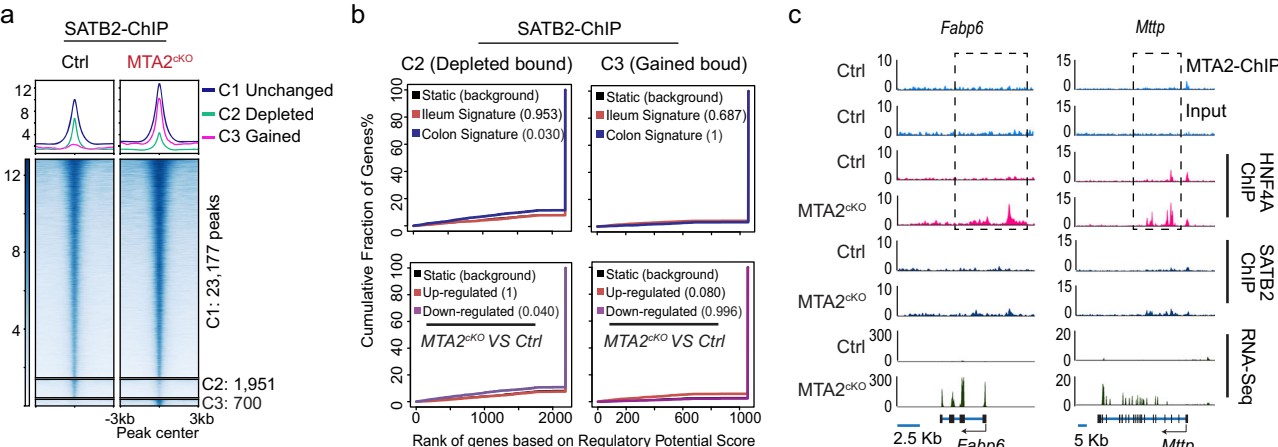

**Fig. 7 | SATB2 genomic binding is affected minimally by *Mta2* loss. a** DNA binding profiles of SATB2 that were unchanged (C1, 23,177 sites), reduced (C2, 1951 sites), or gained (C3, 700 sites) in MTA2cKO colon. Plots are 6-kb windows centered on each HNF4A binding site. **b** Predictions of enhancer regulatory functions by BETA (Binding and Expression Target Analysis) indicate that loss or gain of SATB2 binding in Mta2cKO colon was not associated with transcriptional changes of colonic or ileal genes. Plots depict the cumulative score of regulatory potential for every gene based on enhancer distances from the TSS. Black lines represent the background of unaltered genes, and *p* values denote the significance of positive or negative associations relative to the background. The *P* value calculated using BETA (Binding and Expression Target Analysis) which used one tail KS-test. **c** Genome Browser tracks showed that MTA2 regulated HNF4A genomic binding and transcriptional activation of two small intestine genes (*Fabp6* and *Mttp*) without affecting SATB2 binding.

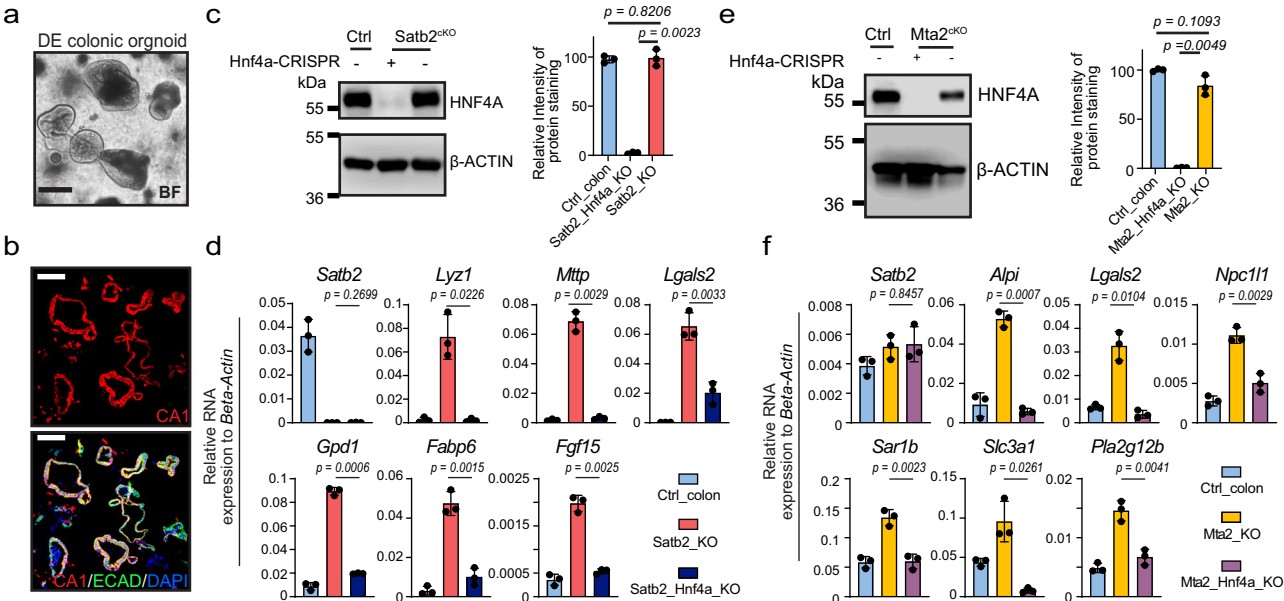

**Fig. 8 | Activation of small intestine genes in Mta2cKO and Satb2cKO colon depends on HNF4A. a, b** Bright field and immunofluorescence pictures of differentiated (DE) colonic organoids. CA1 staining showed enrichment of colonocytes in these organoids. BF: bright field. Three independent experiments were repeated with similar results. Scale bar = 100 μm. **c** Immunoblot and quantification showed that HNF4A levels were comparable in wild-type (WT) vs. Satb2cKO colonoids. CRISPR reduced HNF4A in Satb2/Hnf4a double knockout colonoids to less than 5% of control levels. Mean ± S.D. *n* = 3. *P* value by Unpaired *t*-test, two-tailed. All the different gels/blots were derived from the same experiment and were processed in parallel. **d** QPCR showed that loss of HNF4A blocked small intestine gene activation in *Satb2* mutant colonoids. Mean ± S.D. *n* = 3 independent mice. P value by Unpaired *t*-test, two-tailed. **e** Immunoblot and quantification showed that HNF4A levels were comparable in wild-type (WT) vs. Mta2cKO colonoids. CRISPR reduced HNF4A in Mta2cKO colonoids to less than 2% of control levels. Mean ± S.D. *n* = 3 independent mice. *P* value by Unpaired t-test, two-tailed. All the different gels/blots were derived from the same experiment and were processed in parallel. **f** Loss of HNF4A attenuated small intestine gene activation in *Mta2* mutant colonoids. *Satb2* mRNA levels were not altered by *Hnf4a* deletion. Mean ± S.D. *n* = 3 independent mice. *P* value by Unpaired *t*-test, two-tailed.

(Supplementary Table 2) and cloned into a LentiCRISPRv2 vector (Addgene plasmid #52961). The lentiviruses were packaged with second-generation helper plasmids by transfection with lipofectamine 3000. Each virus titer was determined by counting puromycin-resistant clones in HEK293FT cells 5 days after infection.

To generate the colonoids with gene ablation, a total of $10^5$ cells suspensions (TrypLE digested small colonic fragments with 1 to10 cells per fragment) in 200 μL WENR with 10 μg/mL polybrene were mixed with 20 μL of $10^8$ TCID$_{50}$/ml of virus in one well of a non-tissue culture treated 24 well plate, and centrifuged at 1100 *g* at 37 °C for 30 mins to facilitate infection. After centrifugation, 200 μL of WENR was added and the plate was further incubated for 4 hours at 37 °C. Cells were then resuspended, pelleted, and embedded in Matrigel™ as described in colonoids culture method section. Puromycin selection (1 μg/mL)

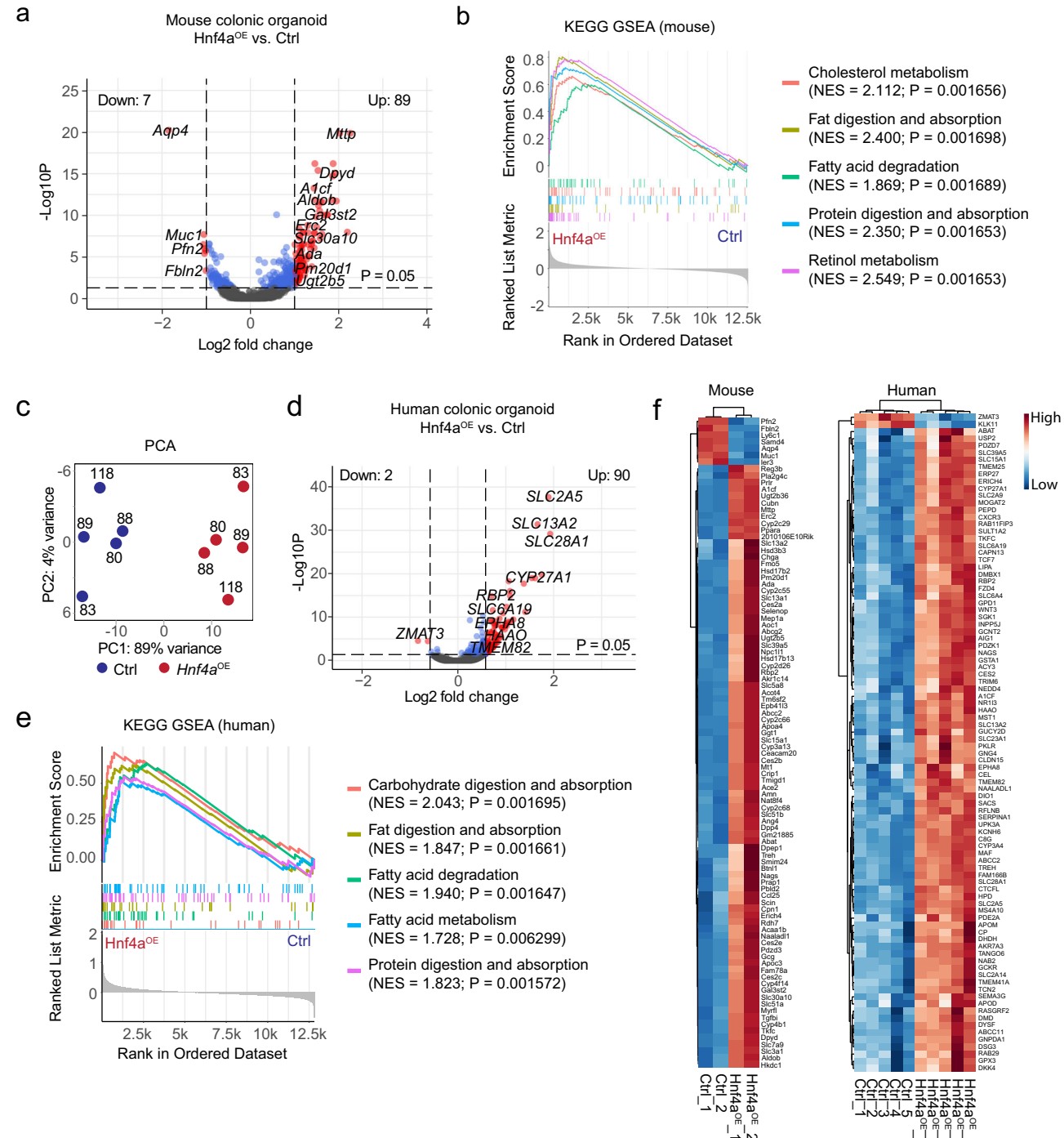

**Fig. 9 | An increase in the level of HNF4A in the colon activates the transcription of small intestine genes.** RNA-sequencing showed over-expression of HNF4A (Hnf4a$^{OE}$) in mouse colonic organoids led to predominant up-regulation of small intestine genes (**a**, volcano plot) enriched for pathways characteristic of small intestine functions as shown in the GSEA plot. P value calculated by (**a**) Wald test or (**b**) phenotype-based permutation test and adjusted by Benjamini-Hochberg method (**a** and **b**). **c** Principal component analysis (PCA) of human colonic organoid transcriptomes from control and HNF4A over-expression (Hnf4a$^{OE}$) samples.

Numbers denote human organoid lines used. RNA-sequencing showed predominant up-regulation of small intestine genes (**d**, volcano plot) in human colonic organoids over-expressing HNF4A. The up-regulated genes were enriched for small intestine functional pathway (**e**). P value calculated by (**d**) Wald test or (**e**) phenotype-based permutation test and adjusted by Benjamini-Hochberg method (**d** and **e**). **f** Heatmaps of all differentially expressed genes (LFC > 1, P < 0.05) in Hnf4a$^{OE}$ murine colonic organoids (left panel) or human organoids (right panel). P value by Wald test and adjusted by Benjamini-Hochberg method.

was initiated 3 days post-infection and lasted for 4 days. After puromycin selection, colonic organoids were sub-cultured into new Matrigel drops and differentiated in DEM for 3 days. The CRISPR-mediated deletion efficiency was analyzed with immunoblotting by using specific target antibodies (Supplementary Data 6).

**Affinity Purification Mass Spectrometry (AP-MS)**
All the experiments were performed on ice or at 4 °C unless specified. Murine proximal colon tissues (half of colon length, about 40 mm) were flushed clean by cold PBS and cut into approximately 3 mm size pieces. The tissues were incubated with 10 mM EDTA/PBS

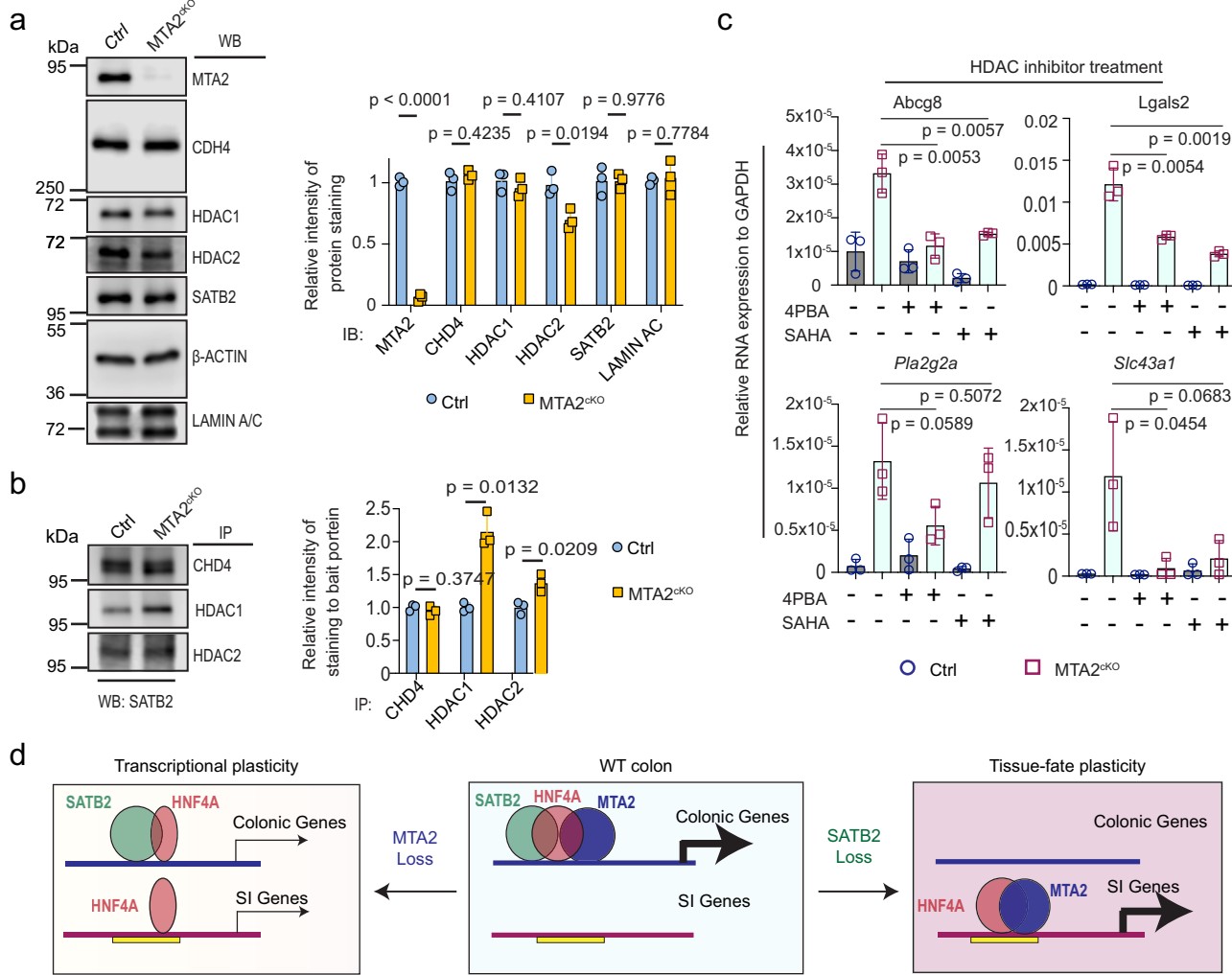

**Fig. 10 | HDAC1/2 interact with SATB2 more strongly in MTA2^cKO vs. control colonoids, and HDAC activity is required for small intestine gene activation in MTA2^cKO colon. a** immunoblots and quantification of SATB2 and core NuRD subunits showed a slight decrease of HDAC2, but no change in HDAC1 or CHD4 in MTA2^cKO vs. control colonoids. Mean ± S.D. *n* = 3 independent samples. *P* value by unpaired t test with Welch correction, adjusted by FDR (1%). All the different gels/blots were derived from the same experiment and were processed in parallel. **b** Co-IP and quantification showed stronger interaction of HDAC1/2 with SATB2, but not CHD4 with SATB2 in MTA2^cKO vs. control colonoids. Mean ± S.D. *n* = 3 independent samples. *P* value by unpaired *t*-test with Welch correction, adjusted by FDR (1%). All the different gels/blots were derived from the same experiment and were processed in parallel. **c** QPCR showed treatment with the HDAC inhibitors 4PBA and SAHA attenuated small intestine gene activation in MTA2^cKO organoids. Mean ± S.D. *n* = 3. *P* value by Unpaired *t*-test. **d** A proposed model of colonocyte plasticity regulation in which MTA2 and SATB2 form a chromatin complex at colonic chromatin to retain HNF4A. SATB2 restrains HNF4A more tightly than MTA2. MTA2 loss leads to a modest depletion of HNF4A on colonic and gain on small intestine (SI) chromatin, and modest down- and up-regulation of colonic and small intestine (SI) genes. In contrast, SATB2 loss results in the untethering of large numbers of HNF4A and consequently transcriptomic shift from colon to small intestine. Yellow bar denotes primed small intestinal enhancers in colon.

in 50 mL tube for 30 min and then transferred to 20 mL cold PBS. After vigorous pipetting with 1% BSA pre-coated 10 mL serological pipettes, the colon tissues were removed to a new tube containing 10 mM EDTA/PBS and incubated for another 30 mins. Epithelium glands in suspensions were collected by centrifugation at 300 g for 5 min as fraction one, followed by resuspending in 10 mL PBS containing 10 μM Y-27632. The second fraction of epithelium glands was collected in the same procedure as fraction one. Two fractions of EDTA-stripped epithelium glands were combined, pelleted, and cross-linked with 2 mM disuccinimidyl glutarate (DSG, Thermo Fisher Scientific, 20593) in PBS at room temperature (RT) for 45 mins. Pellets of epithelial cells were incubated with 0.3 M RIPA buffer (Supplementary Table 3) supplied with Protease Inhibitor Cocktails and sonicated at 20% amplification for 1 min (20 sec on and 20 sec off, 3 cycles). After 10 mins of centrifugation at 18,000 g, supernatants were collected and incubated with anti-SATB2

(Supplementary Data 6) overnight with a rotation speed of 10 RPM. After adding 30 μl protein A/G magnetic beads for 90 mins on the next day, the protein and beads complex was pulled down by a magnetic stander. Total 6 wash of 0.3 M RIPA buffer was performed. Then the DSG-crosslinked SATB2 interaction protein complexes were cleaved from Protein A/G beads by boiling in Laemmli Sample Buffer from Bio-Rad without adding reducing reagents (DTT or 2-Mercaptoethanol) and separated by SDS-PAGE. Proteins in gel were visualized with sliver staining kit from Sigma. The gels over 100 KDa (SATB2 molecular weight) for each sample were cut and digested with trypsin, followed by desalting and LC-MS/MS for protein identification and quantification. The data were processed by MaxQuant and searched against Uniport mouse database with a 1% false discovery rate. Significantly enriched genes were filtered by the quadrilaterals that both samples have SATB2 AP-MS signal intensity and MS count over IgG controls >= 2 fold changes.

## Dual Cross-linking ChIP-Seq for transcription factors

ChIP for Transcription Factors (TFs) SATB2, HNF4A, and MTA2, was performed as described[45]. EDTA-stripped primary colon glands from Ctrl, MTA2cKO, and SATB2cKO were pelleted and cross-linked with 2 mM DSG at RT for 45 mins, followed by 1% formaldehyde fixation for 10 mins. For each experiment, about 50 µl of pelleted cross-linked cells were resuspended in 350 µl sarkosyl lysis buffer (0.25% sarkosyl, 1 mM DTT and Protease Inhibitor Cocktails in 0.3 M RIPA buffer and sonicated at 60% amplification by a tip sonicator (125 W) for 6 min (45 sec on and 45 sec off, 8 cycles) to obtain 200 bp to 800 bp chromatin fragments. Lysates were spun down at 20,000 g at 4 °C to remove insoluble fractions. The supernatant was further diluted in 0.3 M RIPA buffer with Protease Inhibitor Cocktails in a final 2 ml volume. Diluted lysates were incubated with TFs antibodies (Supplementary Data 6) at 4 °C overnight and were additionally incubated with 30 µl protein A/G magnetic beads for 90 mins the next day. A total of 6 washes with cold 0.3 M RIPA buffer and a final rinse with TE buffer (PH 7.5) were performed to remove nonspecific binding. Dual cross-links were reversed overnight by incubating at 65 °C in 1% SDS and 0.1 M NaHCO₃. Any remaining proteins were digested by Proteinase K for 1 hour at 37 °C. Pulled-down genomic DNA was purified with a MinElute purification kit and quantitated by Qubit. The libraries were prepared using the ThruPLEX DNA-Seq Kit, followed by a size selection and purification (200 bp to 800 bp, including index) with AMPure XP beads. The final libraries were quality controlled and pooled for sequencing.

## ChIP-Seq analyses

ChIP-seq libraries (MTA2, HNF4A, and SATB2) were sequenced on an Illumina 4000 instrument to obtain 50-bp pair-end reads. All reads were trimmed by Trim Galore! (0.6.5) (https://www.bioinformatics.babraham.ac.uk/projects/trim_galore/), and subject to quality control with FastQC (0.12.0) before and after adapter trimming. Briefly, Reads were mapped using Bowtie2[46] (2.4.1) to mm10 genome; Peaks were called using callpeak function in the MACS2[47] package (2.2.7.1) with the following parameters (callpeak -t ChIPfile.bam -c Inputfile.bam --format Paired-end BAM --gsize M.musculs(1.87e9) -m 10 30 -q 0.001-bw 300); Bam files were converted into signal files (bigWig) using deep-Tools v3.4.3[46,48]. Signals (bigWig) across samples were quantile normalized with haystack v 0.5.5[49] using 50-bp window across the genome to visualize read distribution on Integrated Genomics Viewer v 2.16.2[50]; BETA -p ChIP_file.bed -e gene_exp.diff -g mm10 -d 50000. We used peaks showing q-val <0.001 and mfold enrichment 10 to 30 and used ChIP-seq from Inputs or knockouts as controls. BETA (1.0.7) was used to associate genes with HNF4A-ChIP depleted or gained bound sites and quantify these associations using peaks within 50-kb from TSS, a significance threshold of FDR-adjusted $P < 0.01$ for differential gene expression in wild-type ileum vs. colon or Mta2cKO vs. Ctrl colon, and other default parameters.

## H3K27ac ChIP-seq

Colon was harvested from euthanized mice and rinsed in PBS. To isolate epithelial cells, colon pieces were rotated in 5 mM EDTA (pH 8.0) in PBS at 4 °C for 30 min with vigorous shaking manually every 10 min. Isolated epithelial cells were fixed at room temperature in 1% formaldehyde (Sigma, F8775) for 15 min. ~1 × 10⁶ cells were lysed in ChIP sonication buffer (50 mM Tris-HCl, 0.1% SDS, 10 mM EDTA, and 1x Roche EDTA-free protease inhibitor) and chromatin was sheared in a Covaris E210 sonicator at 4 °C for 50 min with 5 min on/off cycles to obtain 200−00 bp DNA fragments. Sheared chromatin was incubated overnight with H3K27ac antibody (Active Motif, 39135) at 4 °C. Antibody-bound chromatin complex was incubated with a mixture of 15 µl Protein A and 15 µl Protein G magnetic beads (Thermo Fisher, 10002D and 10004D) for 4 h at 4 °C, washed sequentially twice in low-salt buffer (20 mM Tris-HCl pH 8.1, 2 mM EDTA, 150 mM NaCl, 0.1% SDS, 1% Triton X-100), once in high-salt buffer (20 mM Tris-HCl pH 8.1, 2 mM EDTA, 0.5 M NaCl, 0.1% SDS, 1% Triton X-100), followed by lithium chloride buffer (10 mM Tris-HCl pH 8.1, 1 mM EDTA, 0.25 M LiCl, 1% IGEPAL and 1% deoxycholic acid) and TE buffer (10 mM Tris-HCl pH 8.1, 1 mM EDTA). Chromatin-Antibody complexes were eluted using 100 µl elution buffer (0.1 M NaHCO₃, 1% SDS). Cross-links were reversed using 5 M NaCl solution for 6 h at 65 °C in, samples were treated with proteinase K (Thermo Fisher Scientific, 26160) for 1 h at 55 °C, and DNA was purified using QIAQuick PCR purification kits. Libraries were prepared using ThruPLEX DNA-seq kits (Rubicon Genomics, R400427), and 150 bp pair-end reads were sequenced using Novogene services.

## Edu labeling and Immunostaining

A Click-iT™ EDU Cell Proliferation Kit with Alexa Fluor® 555 (C10338) was used to evaluate proliferation. Briefly, 50 µg EDU per gram of mice bodyweight were intraperitoneal injected for 24 hrs. Mice were euthanized and then intestinal tissues were harvested and flushed clean with cold PBS. The tissues were cut into 1 cm pieces and fixed with 4% paraformaldehyde immediately at 4 °C for 1 hour (Organoids were fixed for 20 mins). After washing with PBS, the tissues were dehydrated by 30% sucrose solution overnight and embedded in O.C.T for a Cryostat sectioning.

Immunofluorescence was performed using a standard procedure, incubating with primary antibodies (Supplementary Data 6) at 4 °C overnight, followed with secondary antibodies at RT for 1 hr. The images were captured using either a confocal microscope (710 Meta) or a Nikon fluorescence microscope. For immunohistochemistry, samples were processed through heat mediated antigen retrieval in Citric Acid buffer (pH 6.0) for 15 mins. Samples were then stained with Anti-MTTP or MTA2 antibodies, followed by Goat anti-Rabbit HRP incubation at RT for 1 hr, and finally, developed with DAB (Brown color, Vector Laboratories, SK-4103) HRP Substrate. Alkaline phosphatase enzyme was detected by Stemgent AP staining kit 2 (Pink color). An Alcian Blue Stain Kit (Vector Laboratories, H-3501) was used to stain goblet cells.

## Western Blot

For Western blot analysis, monoclonal rabbit anti-SATB2 and SMARCD2, monoclonal mouse anti V5 and FLAG, polyclonal rabbit anti-CHD4, MTA2, CTBP2, SMARCA4, SMARCA5, GATAD2A, HDAC1, and HDAC2 antibodies were used to bind target protein, followed by an incubation with a secondary anti-Rabbit Peroxidase (HRP) or anti-Mouse HRP. Protein bands were visualized using enhanced chemiluminescent substrate (Pico from Thermo fisher) and recorded by a Li-COR C-Digit blot scanner. The relative signal intensity was quantified by Image J (v1.51 (100)).

## Bulk RNA sequencing analysis

Bulk RNA-seq was performed as previously described except for mapping to the mouse (mm10) or human (hg38) reference genome respectively[51]. Briefly, reads alignment was performed by STAR (2.7.5b) package[52]. The raw count tables were generated by featureCounts (2.0.1)[53]. The DESeq2 package was used for differential expression analysis[54]. In DESeq2 (1.28.1), the p-values attained by the Wald test are corrected for multiple testing using the Benjamini and Hochberg (BH) method. The Limma (3.44.3) package[55] was used to remove donor-donor variance and batch-effect. Differentially expressed genes were generally determined using parameters of adjusted p-value < 0.05 and LFC > 1 or < −1 unless specified in figure legends. The heatmaps were plotted using the R package, pheatmap (1.0.12). GSEA KEGG analysis and GSEA analysis were conducted with the clusterProfiler package (4.8.1)[56].

## BODIPY staining

Both Mta2cKO and Ctrl mice were under high-fat diet treatment for 3 weeks. The mice were euthanized, about 2 cm size of colon segments

were cut open and flushed with cold DPBS. Colon segments were then incubated with (10 μg/mL) BODIPY in DPBS at RT for 30 mins and followed with 3 times DPBS wash to remove excess BODIPY. Tissues were fixed with 4% paraformaldehyde immediately at 4 °C for 1 hour. After washing with DPBS, the tissues were dehydrated by 30% sucrose solution at 4 °C for 3 hours and embedded in O.C.T for a Cryostat sectioning.

## SATB2 domain deletion

pENTR-3xFlag-3xHA-mSATB2 and overlap PCR primers information were provided in Supplementary Table 2. Each domain deletion PCR amplification was performed by TAKARA-HIFI Tag by following the manufactory guides. PCR fragments were then treated with Dpn1 at 37 °C for 1 hour to remove plasmid template. After gel purified, the fragments were further phosphorylated by T4 PNK and self-ligated by Quick ligase. Each domain deletion cloning vector was confirmed by sequence analysis and recombined into a Dox-inducible Destination vector (Plx403) using Gateway LR Clonase II enzyme kit.

pENTR-mMTA2-V5 was cloned and recombined into Pinducer20. Plx403-SATB2 or each domain deletion expression vector was co-transfected with Pinducer20-MTA2 into HEK293FT cells for 48 hours. Doxycycline (1 μg/mL) was added at 8 hours after transfection to trigger target gene expression.

## Culture of mouse and human colonic organoids and Lentiviral overexpression of HNF4A

The lentiviral backbone plasmid N174-MCS-Puro was a gift from Adam Karpf (Addgene plasmid # 81068). Murine Hnf4a CDS was cloned into the backbone for constitutive expression. The culture of mouse and human primary colon organoids was described previously[21]. For HNF4A overexpression, cultured organoids were dissociated with TrypLE for 3 min at 37 °C and pipetted thoroughly to disperse the organoids into single cells or small clusters (2-3 cells). The dissociated cells were then mixed with appropriate control lentivirus (pLenti-EF1a-Puro-2a-mCherry) or lentivirus overexpressing HNF4A (pLenti-EF1a-Hnf4a-2a-Puro-2a-mCherry) at MOI of 10 and spinfection was performed at 37 °C, 1000 g for 30 min. The cell-virus mixture was further incubated for another 2.5 hours at 37 °C and the cells were then embedded into 3D Matrigel domes. 72 hours after virus infection, puromycin (1ug/mL) was added to culture medium for selection. The selected organoids were differentiated for another 3-5 days and cells were then collected for RNA profiling. All studies involving human samples were approved by ethnic committees at Weill Cornell Medical College.

## Reporting summary

Further information on research design is available in the Nature Portfolio Reporting Summary linked to this article.

## Data availability

The high-throughput sequencing raw and processed data have been deposited to Gene Expression Omnibus (GEO). Bulk RNA-Seq: GSE213879, GSE213878 and GSE245288. ChIP-Seq: GSE213877 and GSE245751. We also analyzed our previously published GEO datasets: GSE148690, GSE167283, GSE167287, and GSE167284. Human (hg38) or mouse (mm10) reference genome sequences used in our study can be found at Gencode GRCh38_v29 or GRCm38_vM20 respectively. All data are available in the main text or the supplementary materials. Source data are provided with this paper.

## Code availability

Code for the data analysis is available at https://github.com/stevehxf/MTA2_NC2024.

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

## Acknowledgements

We thank the WCM Epigenetics Core (Jenny Xiang) for excellent technical assistance. We are grateful to Sean Houghton and Shariq Madha for assistance with computational analyses, Drs. Robert Roeder and Yi Zhang for providing Mta2[f/f] mice. This work was supported by a Tri-Institutional Stem Cell grant to Q.Z and a NIH/NIDDK grant to Q.Z and R.S. (1R01DK125817).

## Author contributions

W.G. and Q.Z devised the experiments, interpreted data, and wrote the manuscript. W.G. performed a majority of the experiments with the help of X.H., Y.L., S.L., M.D., and J.M.G. X.H. analyzed the RNA-seq data and mass spectrometry data. P.N.P.S. analyzed the ChIP-seq data. S.R. and M.V. provided advice and data. L.L. edited the manuscript. R.S. advised on the project and edited the manuscript. Q.Z. supervised the project.

## Competing interests

The authors declare no competing interests.

## Ethics

Human organoids were obtained from the In Vivo Animal and Human Studies Core at the University of Michigan Center for Gastrointestinal Research. All studies involving human samples were approved by the ethics committee at the University of Michigan. Informed consent was obtained from the participants and or parents/guardians for these studies. The research work complied with all relevant ethical regulations. All procedures involving mice were conducted under the IACUC protocol 2018-0050 approved by Weill Cornell Medical College.
