## [Peer Review File · Nature Communications]

A MTA2-SATB2 chromatin complex restrains colonic plasticity toward small intestine by retaining HNF4A at colonic chromatinREVIEWER COMMENTS

Reviewer #1 (Remarks to the Author):

In this study by Gu, et al., the authors performed follow up studies to their previously published manuscript in Cell Stem Cell. In the previous study, the authors found an extremely interesting role of SATB2 in controlling differential enhancer activity in the colon and ileum. This is important because, while there are significant similarities in gene expression in both portions of the intestine, there are fundamental differences in their physiological functions. Thus, reprogramming of cells can potentially have a significant influence on pathologies such as inflammatory bowel disease and colorectal cancer. In the previous study the authors used both in vivo (tissue-specific SATB2 knockout) and ex vivo (organoid) systems to convincingly demonstrate that the identity-defining transcription factor HNF4A binds to different genomic locations in the presence or absence of SATB2. In the current study, they expand these findings by performing mass spectrometry analyses to identify interaction partners of SATB2. One protein identified was MTA2, which is part of the NuRD chromatin remodeling complex. Further studies further explore the role of MTA2 in controlling intestinal cellular identity. Ultimately, the data show that MTA2 loss essentially mimics the effects of SATB2 loss, but to a much lesser degree.

Overall, the experiments are extremely well performed and very convincing. Unfortunately, given the fact that the findings of the study are essentially the same as the previously published manuscript, just with MTA2 now instead of SATB2, I'm afraid the manuscript lacks the novelty necessary for publication in a higher end journal such as Nature Communications. If the authors had additional mechanistic insight into how MTA2 and SATB2 differentially regulate HNF4A localization, this would dramatically help. However, this point does not appear trivial. Specific critique points:

1. As stated above, for publication in Nature Communications, it would be essential that the manuscript provides significant new insights into the molecular mechanisms behind the relocalization of HNF4A following MTA2 (or SATB2) loss. The previous manuscript very nicely demonstrated the differential localization already (following SATB2 loss) such that the only remaining novelty in this manuscript is the finding that MTA2 does the same thing, albeit to a different degree. Do the authors have any further insights from their mass spec data that might provide a hint in this direction? If so, specific experiments could be suggested to attack this aspect.
2. A bit related to the first point. Have the authors performed motif analyses and/or enrichment analyses to determine if the colon- and ileal-specific HNF4A-occupied regions (differentially bound after SATB2 or MTA2 loss) display any differences? Does loss of SATB2 or MTA2 affect interaction of HNF4A with other transcription factors (e.g., but affecting complex formation, or perhaps by affecting the expression of complex partners)? If so, this could provide significant insight into the mechanisms controlling the observed cellular re-programming.
3. A more minor point. Overall, the manuscript is well written. However, it does have minor grammatical errors throughout. These should be corrected before publication here or elsewhere.

Reviewer #2 (Remarks to the Author):

Here Gu et al. report the role of MTA2, a crucial component of the Nucleosome Remodeling Deacetylase complex, in restraining the colonic plasticity of mature colonocytes. Based on their previous work on the function of SATB2 in regulating colonic plasticity, they identified SATB2-interacting proteins in mouse colonic glands and found that multiple NuRD complex components, including MTA2, interacted with SATB2. Disruption of NuRD components in murine colonic organoids resulted in altered expression profiles consistent with those induced by loss of SATB2. Disruption of the NuRD component in murine colonic organoids increases plasticity. Mta2 colon-null mice also showed upregulated expression of lipid absorption genes and increased lipid uptake

capacity. The authors explored the mechanism and found that SATB2, MTA2, and a critical pan-intestinal transcription factor HNF4A co-localized, and the loss of SATB2 or MTA2 altered the binding profile of HNF4A. This study provides some interesting data, which has certain significance for understanding the regulation of colonic genes, especially the function of SATB2. I have two major concerns with this manuscript. One is that the authors propose that the mechanism of action of MTA2 on colonic gene expression is inconsistent with the function of the NuRD complex. NuRD functions by silencing gene expression through its core histone deacetylase activity. But authors argue that MTA2 activates colonic genes. Therefore, it is necessary to analyze the histone acetylation modification near the MTA2 binding site to confirm whether the function of MTA2 depends on the histone deacetylation activity of the NuRD complex. The second is the co-localization of SATB2, MTA2, and HNF4A at colonic enhancers. The authors argued that SATB2 changed the binding profile of HNF4A so that it localizes to the colonic enhancer, so the co-localized region of the three should have significant SATB2 motifs. The prominent HNF4A motif that the authors observed at the MTA2 binding site raise a question. Exactly who binds these enhancers first and then recruits the other two: SATB2 or HNF4A? The author should give an analysis and explanation.

Minor:

1. Fig. 1B The data of AP-MS and the SATB2-interacting protein list are necessary.
2. Fig. 1G and 1H. The gene set used for the GSEA is necessary.
3. Fig. 3C. Authors should provide the differential-expressed gene list.
4. Fig. 4A. Authors should provide the MTA2 associating gene list.
5. Fig. 6. The author does not seem to provide many key data and information for the understanding of this study, including the list of differentially expressed genes, the list of colon genes, MTA2-related genes, etc.

We thank the reviewers for their insightful comments and constructive critiques of the manuscript. In response, we have collected substantial new data and significantly revised the text to clarify key points.

Reviewer #1

In this study by Gu, et al., the authors performed follow up studies to their previously published manuscript in Cell Stem Cell. In the previous study, the authors found an extremely interesting role of SATB2 in controlling differential enhancer activity in the colon and ileum. This is important because, while there are significant similarities in gene expression in both portions of the intestine, there are fundamental differences in their physiological functions. Thus, reprogramming of cells can potentially have a significant influence on pathologies such as inflammatory bowel disease and colorectal cancer. In the previous study the authors used both in vivo (tissue-specific SATB2 knockout) and ex vivo (organoid) systems to convincingly demonstrate that the identity-defining transcription factor HNF4A binds to different genomic locations in the presence or absence of SATB2. In the current study, they expand these findings by performing mass spectrometry analyses to identify interaction partners of SATB2. One protein identified was MTA2, which is part of the NuRD chromatin remodeling complex. Further studies further explore the role of MTA2 in controlling intestinal cellular identity. Ultimately, the data show that MTA2 loss essentially mimics the effects of SATB2 loss, but to a much lesser degree.

Overall, the experiments are extremely well performed and very convincing. Unfortunately, given the fact that the findings of the study are essentially the same as the previously published manuscript, just with MTA2 now instead of SATB2, I'm afraid the manuscript lacks the novelty necessary for publication in a higher end journal such as Nature Communications. If the authors had additional mechanistic insight into how MTA2 and SATB2 differentially regulate HNF4A localization, this would dramatically help. However, this point does not appear trivial.

Specific critique points:

1. As stated above, for publication in Nature Communications, it would be essential that the manuscript provides significant new insights into the molecular mechanisms behind the relocalization of HNF4A following MTA2 (or SATB2) loss. The previous manuscript very nicely demonstrated the differential localization already (following SATB2 loss) such that the only remaining novelty in this manuscript is the finding that MTA2 does the same thing, albeit to a different degree. Do the authors have any further insights from their mass spec data that might provide a hint in this direction? If so, specific experiments could be suggested to attack this aspect.

We agree that this study could benefit from more insight into MTA2's mechanism of action, which we address in the next section. Nonetheless, even in its current form, and especially after the new additions, the study provides two notable insights that advance understanding of colonic transcriptional regulation and plasticity.

1. The involvement of MTA2 in colonic plasticity was never before reported or postulated. Indeed, this is the first study describing the expression pattern and function of MTA2 in normal colon. By uncovering MTA2 via unbiased AP-MS and CRISPR analysis, the study adds an important dimension to the chromatin complex that maintains colonic identity and regulates cellular plasticity.

2. Although we reported HNF4A's movement from colonic to ileal enhancers in *Satb2* mutant colon (PMC8741647), the functional importance of HNF4A in colonic plasticity was not investigated and HNF4A deletion from adult murine colon leads to little transcriptional change (PMC6650150). Therefore, it was not obvious that HNF4A has an important role in colonic plasticity. In this study, using both loss- and gain-of-function analyses, we demonstrate a crucial role for HNF4A in mediating colonic plasticity induced by either *Satb2* loss or *Mta2* loss, revealing a unified molecular mechanism.

2. A bit related to the first point. Have the authors performed motif analyses and/or enrichment analyses to determine if the colon- and ileal-specific HNF4A-occupied regions (differentially bound after SATB2 or MTA2 loss) display any differences? Does loss of SATB2 or MTA2 affect interaction of HNF4A with other transcription factors (e.g., but affecting complex formation, or perhaps by affecting the expression of complex partners)? If so, this could provide significant insight into the mechanisms controlling the observed cellular re-programming.

We appreciate the reviewer's suggestions. We performed a series of experiments to evaluate whether altered SATB2 complex formation in the absence of MTA2 could shed light on the HNF4A relocation and small intestine gene activation in the MTA2 mutant colon.

1. As suggested, we performed motif analysis of colonic and ileal HNF4A-binding regions before and after SATB2 or MTA2 loss, revealing different enriched motifs at these sites (see below). The top motif is always HNF4A, indicating that HNF4A directly engages its canonical binding site in the different conditions.

WT vs *Mta2*^{CKO}

WT enriched

Motif	TFs	Pvalue
	HNF4a	1e-167
	Cdx2	1e-98
	KLF5	1e-75
	Fox:Ebox	1e-54
	ZNF675	1e-25

Mta2^{CKO} enriched

Motif	TFs	Pvalue
	HNF4a	1e-601
	ELF3	1e-100
	IRF1	1e-59
	Smad2::Smad3	1e-52
	CDX4	1e-27

WT vs *Satb2*^{CKO}

WT enriched

Motif	TFs	Pvalue
	HNF4a	1e-269
	CDX4	1e-246
	EKLF	1e-64
	NR2F2	1e-39
	ELF3	1e-30

Satb2^{CKO} enriched

Motif	TFs	Pvalue
	HNF4a	1e-926
	Fra1	1e-204
	NKX6-1	1e-168
	Gata6	1e-135
	Hnf4a	1e-98

2. We then performed Mass Spectrometry to identify SATB2 interacting proteins in MTA2^{CKO} colon, yielding 71 enriched and 25 depleted proteins in MTA2 null vs control colon (Supplementary Table 5). The large number of differential proteins precludes systematic functional studies, but does indicate that MTA2 loss led to substantial changes in the SATB2 complex assembled at colonic enhancers.

3. MTA2 is part of the NuRD complex, whose main effectors are CHD4 and HDAC1/2. Given the observed changes in SATB2 complex after MTA2 loss, we asked whether there could be conformational changes to the SATB2-NuRD. Immunoblots showed unchanged CHD4 and HDAC1 levels, while HDAC2 was modestly reduced in *Mta2*-null colonoids (Fig. 10A). Of note, co-IP detected stronger interactions of HDAC1 and HDAC2 with SATB2, but not CHD4 with SATB2 (Fig. 10B).

4. We hypothesized, based on these findings, that loss of MTA2 alters SATB2-NuRD conformation, bringing HDAC1/2 into closer contact with the SATB2 core complex and leading to enhanced deacetylation, reduced HNF4A binding, and activation of small intestine genes. If so, the prediction is that inhibiting HDAC1/2 would block small intestine gene activation in *Mta2*-null colonic organoids. Indeed, two HDAC1/2 inhibitors SAHA and 4PBA, significantly suppressed activation of small intestine genes including *Abcg8*, *Lgals2*, *Pla2g2a* and *Slc43a1* in *Mta2*-null colonic organoids (Fig. 10C).

Because the substrates of HDAC1/2 include histones H2A, H2B, H3 and H4, we postulated that enhanced HDAC1/2 activity may accelerate removal of H3K27ac at active colonic enhancers. However, ChIP-Seq showed no significant reduction of H3K27ac at loci that were depleted of HNF4A in *Mta2*-mutant colon (Supplementary Fig. 5C). Thus, the primary deacetylation target of NuRD at SATB2 binding sites is not H3K27ac and has yet to be determined.

10C

Supplementary 5C

In summary, our new data suggest a model in which MTA2 loss leads to conformational changes of the SATB2 complex anchored at colonic enhancers (diagram below). We propose that HDAC1/2 come closer to the SATB2 complex after MTA2 loss and that enhanced HDAC1/2 activity near SATB2 weakens HNF4A binding, leading to its relocation from colonic to ileal enhancers and activation of ileal genes. Future studies are needed to fully validate this model.

3. A more minor point. Overall, the manuscript is well written. However, it does have minor grammatical errors throughout. These should be corrected before publication here or elsewhere.

We appreciate this comment and have corrected grammatical errors throughout the revised manuscript.

Reviewer #2

Here Gu et al. report the role of MTA2, a crucial component of the Nucleosome Remodeling Deacetylase complex, in restraining the colonic plasticity of mature colonocytes. Based on their previous work on the function of SATB2 in regulating colonic plasticity, they identified SATB2-interacting proteins in mouse colonic glands and found that multiple NuRD complex components, including MTA2, interacted with SATB2. Disruption of NuRD components in murine colonic organoids resulted in altered expression profiles consistent with those induced by loss of SATB2. Disruption of the NuRD component in murine colonic organoids increases plasticity. Mta2 colon-null mice also showed upregulated expression of lipid absorption genes and increased lipid uptake capacity. The authors explored the mechanism and found that SATB2, MTA2, and a critical pan-intestinal transcription factor HNF4A co-localized, and the loss of SATB2 or MTA2 altered the binding profile of HNF4A. This study provides some interesting data, which has certain significance for understanding the regulation of colonic genes, especially the function of SATB2. I have two major concerns with this manuscript. One is that the authors propose that the mechanism of action of MTA2 on colonic gene expression is inconsistent with the function of the NuRD complex. NuRD functions by silencing gene expression through its core histone deacetylase activity. But authors argue that MTA2 activates colonic genes. Therefore, it is necessary to analyze the histone acetylation modification near the MTA2 binding site to confirm whether the function of MTA2 depends on the histone deacetylation activity of the NuRD complex.

We agree that the canonical function of NuRD is transcriptional silencing. However, our ChIP-seq data reveal that MTA2-NuRD binding concentrates at active colonic enhancers. Indeed, NuRD has been detected at active enhancers in many tissues (PMC6039721, PMC5385134).

Following the reviewer's suggestion, we examined histone (H3K27) acetylation at MTA2 binding sites in WT and *Mta2*^{CKO} mouse colon, using ChIP-seq. We detected no overall difference in H3K27ac signals (Supplementary Fig 5A). We then focused our analysis on HNF4A binding sites that were depleted or gained in *Mta2*^{CKO} mutant colon. Depleted sites showed no significant H3K27ac loss (Supplementary Fig 5B) whereas gained sites showed increased H3K27ac signals, consistent with the activation of small intestine enhancers in *Mta2* mutant (Supplementary Fig 5C). These data suggest that MTA2-NuRD does not directly deacetylate H3K27 at colonic binding sites. These findings are consistent with observations in other systems (PMC4793962) that HDACs have broad substrates including H2A, H2B, H3 and H4, and other non-histone proteins.

Supplementary 5A

Supplementary 5B

Supplementary 5C

We then hypothesized that loss of MTA2 may alter HDAC1/2 positioning within the SATB2 complex and weaken HNF4A binding at colonic enhancers, leading to its relocation to small intestine enhancers. Indeed, co-IP studies showed stronger interactions of HDAC1 and HDAC2 with SATB2 in *Mta2*-null vs control colonic organoids (Fig. 10B). Moreover, HDAC1/2 inhibitors SAHA and 4PBA suppressed activation of small intestine genes including *Abcg8*, *Lgals2*, *Pla2g2a* and *Slc43a1* in *Mta2*-null colonic organoids (Fig. 10C).

Fig 10B

Fig 10C

Fig 10B

Our new data further contribute to a model in which MTA2 loss leads to conformational changes of the SATB2 complex anchored at colonic enhancers. We propose that HDAC1/2 are brought closer to the SATB2 complex in the absence of MTA2 and interact more strongly with SATB2. This enhanced HDAC1/2 activity near SATB2 weakens HNF4A binding and enables its relocation from colonic to ileal enhancers, hence activating ileal genes. This model is consistent with the classical view that HDAC1/2 plays a suppressive function.

The second is the co-localization of SATB2, MTA2, and HNF4A at colonic enhancers. The authors argued that SATB2 changed the binding profile of HNF4A so that it localizes to the colonic enhancer, so the co-localized region of the three should have significant SATB2 motifs. The prominent HNF4A motif that the authors observed at the MTA2 binding site raise a question. Exactly who binds these enhancers first and then recruits the other two: SATB2 or HNF4A? The author should give an analysis and explanation.

Our data reveal prominent co-localization of SATB2, MTA2, and HNF4A at colonic enhancers. Data presented in the original submission also showed that upon SATB2 loss, both MTA2 and HNF4A moved from colonic to small intestine enhancers. The revised manuscript includes ChIP-Seq of SATB2 in

Mta2-null vs control colon, revealing that few SATB2 binding sites are depleted (1,951) or gained (700) in *MTA2^{CKO}* colon (Fig. 7A). Our analysis of the regulatory potential of these sites suggests that they do not materially regulate transcription (Fig. 7B). Thus, in contrast to SATB2 loss, MTA2 loss does not significantly influence SATB2 genomic binding or its transcriptional regulation.

Fig. 7A

Fig. 7B

Deletion of *Hnf4a* does not alter intestinal transcription because its homologue, *Hnf4g*, is up-regulated and binds at the same genomic sites; deletion of both *Hnf4a* and *Hnf4g* caused rapid intestine degeneration and death (PMC6650150). At this point, we do not know if there is a temporal sequence for genomic binding of SATB2, MTA2 and HNF4A, but our data indicate that SATB2 is the linchpin in colonic transcriptional regulation. Genomic binding of MTA2 and HNF4A is regulated by SATB2 but not vice versa.

Minor:

1. Fig. 1B The data of AP-MS and the SATB2-interacting protein list are necessary.

We now present all proteins in SATB2 AP-MS in Supplementary Table 1.

2. Fig. 1G and 1H. The gene set used for the GSEA is necessary.

The gene set used in Figures 1G and 1H is now shown in Supplementary Table 2.

3. Fig. 3C. Authors should provide the differential-expressed gene list.

The complete DEG list associated with Fig. 3C is now shown in Supplementary Table 3.

4. Fig. 4A. Authors should provide the MTA2 associating gene list.

4. Genes proximal to MTA2 binding sites in colon (< 50 kb) are now listed in Supplementary Table 4.

5. Fig. 6. The author does not seem to provide many key data and information for the understanding of this study, including the list of differentially expressed genes, the list of colon genes, MTA2-related genes, etc.

The original Figure 6 contained both loss- and gain-of-function of HNF4A in colonic organoids. To provide additional data and description, we have separated the data into two figures, one focusing on LOF (Figure 8) and the other on GOF (Figure 9). We include a heatmap of all genes dysregulated in HNF4A GOF studies in murine (original submission) and human (this revision) colonic organoids (Fig. 9F). Data from the two species are concordant, indicating conserved HNF4A functions in colonic transcription.

Fig 9

REVIEWERS' COMMENTS

Reviewer #1 (Remarks to the Author):

Overall a very beautiful, well performed study. While I would have liked to have seen more mechanistic and deeper evaluation of the differential HNF4A binding sites with or without SATB2 or MTA2, the authors have already a very extensive and solid body of data. Thus, I do not feel it would be sensible to further slow down publication by requiring additional studies. Thus, I recommend acceptance of the manuscript for publication.

Reviewer #2 (Remarks to the Author):

The manuscript discusses the role of MTA2, a component of the NuRD complex, in regulating colonic plasticity and gene expression. The authors provide new data and insights into the mechanisms underlying the regulation of colonic genes, particularly in the context of SATB2 function.

In their revision, the authors have made a significant effort to address the reviewer's concerns. They have provided additional experimental data, including ChIP-Seq analysis of histone acetylation (H3K27ac) and SATB2 binding in the *Mta2*-null colon, which is crucial for understanding the role of MTA2 and NuRD in gene regulation. These data support the authors' claim that MTA2-NuRD does not directly deacetylate H3K27 at colonic binding sites. They also offer a model explaining how MTA2 loss influences SATB2 complex conformation and interactions with HDAC1/2, leading to altered HNF4A binding and subsequent activation of small intestine genes.

Furthermore, the authors have clarified the relationship between SATB2, MTA2, and HNF4A at colonic enhancers, providing evidence that SATB2 is the linchpin in colonic transcriptional regulation and that MTA2 and HNF4A's genomic binding is regulated by SATB2 but not vice versa. This clarification addresses the reviewer's concerns about the order of binding and recruitment among these factors.

The revision effectively addresses the major concerns raised by the reviewer, and the additional data and explanations significantly enhance the manuscript's quality and support the conclusions. The authors have successfully demonstrated the role of MTA2 and NuRD in colonic plasticity and provided valuable insights into regulating colonic genes. Overall, the manuscript has been improved and may be considered for acceptance with these revisions.

We thank the reviewers for their kind comments and encouragement. The reviewers did not raise more questions. We will revise the manuscript according to the editor's requests.

Reviewer #1

Overall a very beautiful, well performed study. While I would have liked to have seen more mechanistic and deeper evaluation of the differential HNF4A binding sites with or without SATB2 or MTA2, the authors have already a very extensive and solid body of data. Thus, I do not feel it would be sensible to further slow down publication by requiring additional studies. Thus, I recommend acceptance of the manuscript for publication.

Reviewer #2

The manuscript discusses the role of MTA2, a component of the NuRD complex, in regulating colonic plasticity and gene expression. The authors provide new data and insights into the mechanisms underlying the regulation of colonic genes, particularly in the context of SATB2 function.

In their revision, the authors have made a significant effort to address the reviewer's concerns. They have provided additional experimental data, including ChIP-Seq analysis of histone acetylation (H3K27ac) and SATB2 binding in the Mta2-null colon, which is crucial for understanding the role of MTA2 and NuRD in gene regulation. These data support the authors' claim that MTA2-NuRD does not directly deacetylate H3K27 at colonic binding sites. They also offer a model explaining how MTA2 loss influences SATB2 complex conformation and interactions with HDAC1/2, leading to altered HNF4A binding and subsequent activation of small intestine genes.

Furthermore, the authors have clarified the relationship between SATB2, MTA2, and HNF4A at colonic enhancers, providing evidence that SATB2 is the linchpin in colonic transcriptional regulation and that MTA2 and HNF4A's genomic binding is regulated by SATB2 but not vice versa. This clarification addresses the reviewer's concerns about the order of binding and recruitment among these factors.

The revision effectively addresses the major concerns raised by the reviewer, and the additional data and explanations significantly enhance the manuscript's quality and support the conclusions. The authors have successfully demonstrated the role of MTA2 and NuRD in colonic plasticity and provided valuable insights into regulating colonic genes. Overall, the manuscript has been improved and may be considered for acceptance with these revisions.